# Offline Reinforcement Learning with Mixture of Deterministic Policies

**Takayuki Osa**                                              *osa@mi.t.u-tokyo.ac.jp*
*The University of Tokyo, RIKEN*

**Akinobu Hayashi**                                      *akinobu_hayashi@jp.honda*
*Honda R&D Co., Ltd.*

**Pranav Deo**                                                 *pranav_deo@jp.honda*
*Honda R&D Co., Ltd.*

**Naoki Morihira**                                          *naoki_morihira@jp.honda*
*Honda R&D Co., Ltd.*

**Takahide Yoshiike**                                    *takahide_yoshiike@jp.honda*
*Honda R&D Co., Ltd.*

**Reviewed on OpenReview:** `https://openreview.net/forum?id=zkRCp4RmAF`

## Abstract

Offline reinforcement learning (RL) has recently attracted considerable attention as an approach for utilizing past experiences to learn a policy. Recent studies have reported the challenges of offline RL, such as estimating the values of actions that are outside the data distribution. To mitigate offline RL issues, we propose an algorithm that leverages a mixture of deterministic policies. When the data distribution is multimodal, fitting a policy modeled with a unimodal distribution, such as Gaussian distribution, may lead to interpolation between separate modes, thereby resulting in the value estimation of actions that are outside the data distribution. In our framework, the state-action space is divided by learning discrete latent variables, and the sub-policies corresponding to each region are trained. The proposed algorithm was derived by considering the variational lower bound of the offline RL objective function. We show empirically that the use of the proposed mixture policy can reduce the accumulation of the critic loss in offline RL, which was reported in previous studies. Experimental results also indicate that using a mixture of deterministic policies in offline RL improves the performance with the D4RL benchmarking datasets.

## 1 Introduction

Reinforcement learning (RL) (Sutton & Barto, 2018) has achieved remarkable success in various applications. Many of its successes have been achieved in online learning settings where the RL agent interacts with the environment during the learning process. However, such interactions are often time-consuming and computationally expensive. The aim of reducing the number of interactions in RL has spurred active interest in offline RL (Levine et al., 2020), also known as batch RL (Lange et al., 2012). In offline RL, the goal is to learn the optimal policy from a prepared dataset generated through arbitrary and unknown processes. Prior work on offline RL has focused on how to avoid estimating the Q-values of actions that are outside the data distribution (Fujimoto et al., 2019; Fujimoto & Gu, 2021). In this study, we propose addressing it from the perspective of the policy structure. Our hypothesis is that, if the data distribution in a given

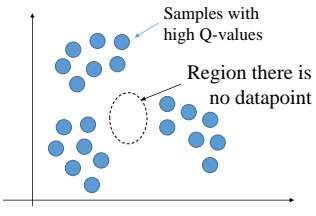

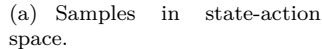

(a) Samples in state-action space.

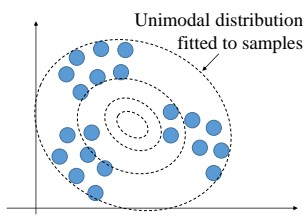

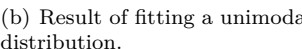

(b) Result of fitting a unimodal distribution.

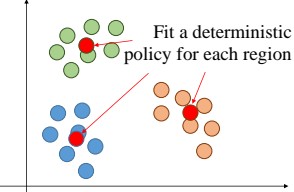

(c) Proposed approach.

Figure 1: Illustration of the proposed approach. (a) In offline RL, the distribution of samples is often multimodal; (b) Fitting a unimodal distribution to such samples can lead to generating the action out of the data distribution; (c) In the proposed approach, first the latent discrete variable of the state-action space is learned, and then a deterministic policy is learned for each region.

dataset is multimodal, the evaluation of the out-of-distribution actions can be reduced by leveraging a policy conditioned on discrete latent variables, which can be interpreted as dividing the state-action space and learning sub-policies for each region. When the data distribution is multimodal, as shown in Figure 1(a), fitting a policy modeled with a unimodal distribution, such as Gaussian distribution, may lead to interpolation between separate modes, thereby resulting in the value estimation of actions that are outside the data distribution (Figure 1(b)). To avoid this, we employ a mixture of deterministic policies (Figure 1(c)). We divide the state-action space by learning discrete latent variables and learn the sub-policies for each region. Ideally, this approach can help avoid interpolating separate modes of the data distribution.

The main contributions of this study are as follows: 1) it provides a practical algorithm for training a mixture of deterministic policies in offline RL and 2) investigates the effect of policy structure in offline RL. Although it is expected that a mixture of deterministic policies has advantages over a monolithic policy, it is not trivial to train a mixture of deterministic policies. We derived the proposed algorithm by considering the variational lower bound of the offline RL objective function. We refer to the proposed algorithm as deterministic mixture policy optimization (DMPO). Additionally, we proposed a regularization technique for a mixture policy based on mutual information. We empirically demonstrate that the proposed regularization technique improves the performance of the proposed algorithm. A previous study (Brandfonbrener et al., 2021) reported the accumulation of critic loss values during the training phase, which was attributed to generating out-of-distribution actions. In our experiments, we investigated the effect of the policy structures in offline RL through comparison with methods that use a monolithic deterministic policy, Gaussian policy, and Gaussian mixture policy. We empirically show that the use of a mixture of deterministic policies can reduce the accumulation of the approximation error in offline RL. Although a mixture of Gaussian policies has been used in the online RL literature, we show that the use of a Gaussian mixture policy does not significantly improve the performance of an offline RL algorithm. Through experiments with benchmark tasks in D4RL (Fu et al., 2020), we demonstrate that the proposed algorithms are competitive with prevalent offline RL methods.

## 2 Related Work

Recent studies have shown that regularization is a crucial component of offline RL (Fujimoto et al., 2019; Kumar et al., 2020; Levine et al., 2020; Kostrikov et al., 2021). For example, Kostrikov et al. proposed a regularization based on Fisher divergence Kostrikov et al. (2021), and Fujimoto et al. showed that simply adding a behavior cloning term to the objective function in TD3 can achieve state-of-the-art performance on D4RL benchmark tasks Fujimoto & Gu (2021). Other studies have investigated the structure of the critic, proposing the use of an ensemble of critics (An et al., 2021) or offering a one-step offline RL approach (Gulcehre et al., 2020; Brandfonbrener et al., 2021; Goo & Niekum, 2021). Previous studies (Fujimoto et al., 2019; Fujimoto & Gu, 2021) have indicated that the source of the value approximation error is "extrapolation error" that occurs when the value of state-action pairs that are not contained in a given dataset is estimated. We hypothesize that such an "extrapolation error" can be mitigated by dividing the state-action space, which

can be potentially achieved by learning discrete latent variables. We investigate the effect of incorporating policy structure as an inductive bias in offline RL, which has not been thoroughly investigated.

Learning the discrete latent variable in the context of RL is closely related to a mixture policy, where a policy is represented as a combination of a finite number of sub-policies. In a mixture policy, one of the sub-policies is activated for a given state, and the module that determines which sub-policy is to be used is often called the gating policy (Daniel et al., 2016). Because of its two-layered structure, a mixture policy is also called *a hierarchical policy* (Daniel et al., 2016). Although we did not consider temporal abstraction in this study, we note that a well-known hierarchical RL framework with temporal abstraction is the option critic (Bacon et al., 2017). Because we consider policies without temporal abstraction, we use the term "mixture policy," following the terminology in Wulfmeier et al. (2021). Previous studies have demonstrated the advantages of mixture policies for online RL (Osa et al., 2019; Zhang & Whiteson, 2019; Wulfmeier et al., 2020; 2021; Akrour et al., 2021). In these existing methods, sub-policies are often trained to cover separate modes of the Q-function, which is similar to our approach. Although existing methods have leveraged the latent variable in offline RL (Zhou et al., 2020; Chen et al., 2021b; 2022), the latent variable is continuous in these methods. For example, Chen et al. recently proposed an algorithm called latent-variable advantage-weighted policy optimization (LAPO), which leverages continuous latent space for policy learning Chen et al. (2022). LAPO incorporates an importance weight based on the advantage function and learns the continuous latent variable. Although LAPO can achieve state-of-the-art performance on well-known benchmark tasks, we empirically show in this study that LAPO suffers from a surge of the critic loss during training.

## 3 Problem Formulation

**Reinforcement Learning** Consider a reinforcement learning problem under a Markov decision process (MDP) defined by a tuple $(\mathcal{S}, \mathcal{A}, \mathcal{P}, r, \gamma, d)$, where $\mathcal{S}$ is the state space, $\mathcal{A}$ is the action space, $\mathcal{P}(\boldsymbol{s}_{t+1} | \boldsymbol{s}_t, \boldsymbol{a}_t)$ is the transition probability density, $r(\boldsymbol{s}, \boldsymbol{a})$ is the reward function, $\gamma$ is the discount factor, and $d(\boldsymbol{s}_0)$ is the probability density of the initial state. A policy $\pi(\boldsymbol{a}|\boldsymbol{s}) : \mathcal{S} \times \mathcal{A} \mapsto \mathbb{R}$ is defined as the conditional probability density over the actions given the states. The goal of RL is to identify a policy that maximizes the expected return $\mathbb{E}[R_0|\pi]$, where the return is the sum of the discounted rewards over time given by $R_t = \sum_{k=t}^{T} \gamma^{k-t} r(\boldsymbol{s}_k, \boldsymbol{a}_k)$. The Q-function, $Q^\pi(\boldsymbol{s}, \boldsymbol{a})$, is defined as the expected return when starting from state $\boldsymbol{s}$ and taking action $\boldsymbol{a}$, then following policy $\pi$ under a given MDP (Sutton & Barto, 2018).

In offline RL, it is assumed that the learning agent is provided with a fixed dataset, $\mathcal{D} = \{(\boldsymbol{s}_i, \boldsymbol{a}_i, r_i)\}_{i=1}^{N}$, comprising states, actions, and rewards collected by an unknown behavior policy. The goal of offline RL is to obtain a policy that maximizes the expected return using $\mathcal{D}$ without online interactions with the environment during the learning process.

**Objective function** We formulate the offline RL problem as follows: given dataset $\mathcal{D} = \{(\boldsymbol{s}_i, \boldsymbol{a}_i, r_i)\}_{i=1}^{N}$ obtained through the interactions between behavior policy $\beta(\boldsymbol{a}|\boldsymbol{s})$ and the environment, our goal is to obtain policy $\pi$ that maximizes the expected return. In the process of training a policy in offline RL, the expected return is evaluated with respect to the states stored in the given dataset. Thus, the objective function is given by:

$$J(\pi) = \mathbb{E}_{\boldsymbol{s} \sim \mathcal{D}, \boldsymbol{a} \sim \pi} \left[ f^\pi(\boldsymbol{s}, \boldsymbol{a}) \right], \tag{1}$$

where $f^\pi$ is a function that quantifies the performance of policy $\pi$. There are several choices for $f^\pi$ as indicated in Schulman et al. (2016). TD3 employed the action-value function, $f^\pi(\boldsymbol{s}, \boldsymbol{a}) = Q^\pi(\boldsymbol{s}, \boldsymbol{a})$, and A2C employed the advantage-function $f^\pi(\boldsymbol{s}, \boldsymbol{a}) = A^\pi(\boldsymbol{s}, \boldsymbol{a})$ (Mnih et al., 2016). Other previous studies employed shaping with an exponential function, such as $f^\pi(\boldsymbol{s}, \boldsymbol{a}) = \exp\left(Q^\pi(\boldsymbol{s}, \boldsymbol{a})\right)$ (Peters & Schaal, 2007) or $f^\pi(\boldsymbol{s}, \boldsymbol{a}) = \exp\left(A^\pi(\boldsymbol{s}, \boldsymbol{a})\right)$ (Neumann & Peters, 2008; Wang et al., 2018). Without a loss of generality, we assume that the objective function is given by Equation 1. We derive the proposed algorithm by considering the lower bound of the objective function of offline RL in Equation 1.

**Mixture policy** In this study, we consider a mixture of policies given by

$$\pi(\boldsymbol{a}|\boldsymbol{s}) = \sum_{\boldsymbol{z} \in \mathcal{Z}} \pi_{\text{gate}}(\boldsymbol{z}|\boldsymbol{s})\pi_{\text{sub}}(\boldsymbol{a}|\boldsymbol{s}, \boldsymbol{z}), \tag{2}$$

where $\boldsymbol{z}$ is a discrete latent variable, $\pi_{\text{gate}}(\boldsymbol{z}|\boldsymbol{s})$ is the gating policy that determines the value of the latent variable, and $\pi_{\text{sub}}(\boldsymbol{a}|\boldsymbol{s}, \boldsymbol{z})$ is the sub-policy that determines the action for a given $\boldsymbol{s}$ and $\boldsymbol{z}$. We assume that a sub-policy $\pi_{\text{sub}}(\boldsymbol{a}|\boldsymbol{s}, \boldsymbol{z})$ is deterministic; the sub-policy determines the action for a given $\boldsymbol{s}$ and $\boldsymbol{z}$ in a deterministic manner as $\boldsymbol{a} = \boldsymbol{\mu_\theta}(\boldsymbol{s}, \boldsymbol{z})$, where $\boldsymbol{\mu_\theta}(\boldsymbol{s}, \boldsymbol{z})$ is parameterized by vector $\boldsymbol{\theta}$. Additionally, we assume that the gating policy $\pi_{\text{gate}}(\boldsymbol{z}|\boldsymbol{s})$ determines the latent variable as:

$$\boldsymbol{z} = \arg\max_{\boldsymbol{z}'} Q_{\boldsymbol{w}}(\boldsymbol{s}, \boldsymbol{\mu_\theta}(\boldsymbol{s}, \boldsymbol{z}')), \tag{3}$$

where $Q_{\boldsymbol{w}}(\boldsymbol{s}, \boldsymbol{a})$ is the estimated Q-function parameterized by vector $\boldsymbol{w}$. This gating policy is applicable to objective functions such as $f^\pi(\boldsymbol{s}, \boldsymbol{a}) = \exp(Q^\pi(\boldsymbol{s}, \boldsymbol{a}))$, $f^\pi(\boldsymbol{s}, \boldsymbol{a}) = A^\pi(\boldsymbol{s}, \boldsymbol{a})$, and $f^\pi(\boldsymbol{s}, \boldsymbol{a}) = \exp(A^\pi(\boldsymbol{s}, \boldsymbol{a}))$. Please refer to Appendix A for details.

## 4 Training a mixture of deterministic policies by maximizing the variational lower bound

We consider a training procedure based on policy iteration (Sutton & Barto, 2018), in which the critic and policy are iteratively improved. In this section, we describe the policy update procedure of the proposed method.

### 4.1 Variational lower bound for offline RL

To derive the update rule for policy parameter $\boldsymbol{\theta}$, we first consider the lower bound of objective function $\log J(\pi)$ in Equation 1. We assume that $f^\pi(\boldsymbol{s}, \boldsymbol{a})$ in Equation 1 is approximated with $\hat{f}_{\boldsymbol{w}}^\pi(\boldsymbol{s}, \boldsymbol{a})$, which is parameterized with a vector $\boldsymbol{w}$. In a manner similar to Dayan & Hinton (1997); Kober & Peters (2011), when $\hat{f}_{\boldsymbol{w}}^\pi(\boldsymbol{s}, \boldsymbol{a}) > 0$ for any $\boldsymbol{s}$ and $\boldsymbol{a}$, we can determine the lower bound of $\log J(\pi)$ using Jensen's inequality as follows:

$$\log J(\pi) \approx \log \int d^\beta(\boldsymbol{s})\pi_{\boldsymbol{\theta}}(\boldsymbol{a}|\boldsymbol{s})\hat{f}_{\boldsymbol{w}}^\pi(\boldsymbol{s}, \boldsymbol{a})d\boldsymbol{s}d\boldsymbol{a} \tag{4}$$

$$= \log \int d^\beta(\boldsymbol{s})\beta(\boldsymbol{a}|\boldsymbol{s})\frac{\pi_{\boldsymbol{\theta}}(\boldsymbol{a}|\boldsymbol{s})}{\beta(\boldsymbol{a}|\boldsymbol{s})}\hat{f}_{\boldsymbol{w}}^\pi(\boldsymbol{s}, \boldsymbol{a})d\boldsymbol{s}d\boldsymbol{a} \tag{5}$$

$$\geq \int d^\beta(\boldsymbol{s})\beta(\boldsymbol{a}|\boldsymbol{s})\log\left(\frac{\pi_{\boldsymbol{\theta}}(\boldsymbol{a}|\boldsymbol{s})}{\beta(\boldsymbol{a}|\boldsymbol{s})}\right)\hat{f}_{\boldsymbol{w}}^\pi(\boldsymbol{s}, \boldsymbol{a})d\boldsymbol{s}d\boldsymbol{a} \tag{6}$$

$$= \mathbb{E}_{(\boldsymbol{s}, \boldsymbol{a}) \sim \mathcal{D}}\left[\log \pi_{\boldsymbol{\theta}}(\boldsymbol{a}|\boldsymbol{s})\hat{f}_{\boldsymbol{w}}^\pi(\boldsymbol{s}, \boldsymbol{a})\right] - \mathbb{E}_{(\boldsymbol{s}, \boldsymbol{a}) \sim \mathcal{D}}\left[\log \beta(\boldsymbol{a}|\boldsymbol{s})\hat{f}_{\boldsymbol{w}}^\pi(\boldsymbol{s}, \boldsymbol{a})\right], \tag{7}$$

where $\beta(\boldsymbol{a}|\boldsymbol{s})$ is the behavior policy used for collecting the given dataset, and $d^\beta(\boldsymbol{s})$ is the stationary distribution over the state induced by executing behavior policy $\beta(\boldsymbol{a}|\boldsymbol{s})$. The second term in Equation 7 is independent of policy parameter $\boldsymbol{\theta}$. Thus, we can maximize the lower bound of $J(\pi)$ by maximizing $\sum_{i=1}^N \log \pi_{\boldsymbol{\theta}}(\boldsymbol{a}_i|\boldsymbol{s}_i)\hat{f}_{\boldsymbol{w}}^\pi(\boldsymbol{s}_i, \boldsymbol{a}_i)$. When we employ $f^\pi(\boldsymbol{s}, \boldsymbol{a}) = \exp(A^\pi(\boldsymbol{s}, \boldsymbol{a}))$, and the policy is Gaussian, the resulting algorithm is equivalent to AWAC (Nair et al., 2020). To employ a mixture policy with a discrete latent variable, we further analyze the objective function in Equation 7. As in Kingma & Welling (2014); Sohn et al. (2015), we obtain a variant of the variational lower bound of the conditional log-likelihood:

$$\log \pi_{\boldsymbol{\theta}}(\boldsymbol{a}_i|\boldsymbol{s}_i) \geq -D_{\text{KL}}(q_{\boldsymbol{\phi}}(\boldsymbol{z}|\boldsymbol{s}_i, \boldsymbol{a}_i)||p(\boldsymbol{z}|\boldsymbol{s}_i)) + \mathbb{E}_{\boldsymbol{z} \sim p(\boldsymbol{z}|\boldsymbol{s}_i, \boldsymbol{a}_i)}\left[\log \pi_{\boldsymbol{\theta}}(\boldsymbol{a}_i|\boldsymbol{s}_i, \boldsymbol{z})\right]$$
$$= \ell_{\text{cvae}}(\boldsymbol{s}_i, \boldsymbol{a}_i; \boldsymbol{\theta}, \boldsymbol{\phi}), \tag{8}$$

where $q_{\boldsymbol{\phi}}(\boldsymbol{z}|\boldsymbol{s}, \boldsymbol{a})$ is the approximate posterior distribution parameterized with vector $\boldsymbol{\phi}$, and $p(\boldsymbol{z}|\boldsymbol{s})$ is the true posterior distribution. The derivation of Equation 8 is provided in Appendix B. Although it is often

assumed in prior studies (Fujimoto et al., 2019) that $\boldsymbol{z}$ is statistically independent of $\boldsymbol{s}$, that is, $p(\boldsymbol{z}|\boldsymbol{s}) = p(\boldsymbol{z})$, in our framework, $p(\boldsymbol{z}|\boldsymbol{s})$ should represent the behavior of the gating policy, $\pi_{\boldsymbol{\theta}}(\boldsymbol{z}|\boldsymbol{s})$. In our framework, the gating policy $\pi_{\text{gate}}(\boldsymbol{z}|\boldsymbol{s})$ determines the latent variable as $\boldsymbol{z} = \arg\max_{\boldsymbol{z}'} Q_{\boldsymbol{w}}(\boldsymbol{s}, \mu(\boldsymbol{s}, \boldsymbol{z}'))$. However, the gating policy is not explicitly modeled in our framework because it would increase computational complexity. To approximate the gating policy represented by the argmax function over the Q-function, we used the softmax distribution, which is often used to approximate the argmax function, given by

$$p(\boldsymbol{z}|\boldsymbol{s}) = \frac{\exp\left(Q_{\boldsymbol{w}}(\boldsymbol{s}, \boldsymbol{\mu}_{\boldsymbol{\theta}}(\boldsymbol{s}, \boldsymbol{z}))\right)}{\sum_{\boldsymbol{z} \in \mathcal{Z}} \exp\left(Q_{\boldsymbol{w}}(\boldsymbol{s}, \boldsymbol{\mu}_{\boldsymbol{\theta}}(\boldsymbol{s}, \boldsymbol{z}))\right)}. \tag{9}$$

Since we employ double-clipped Q-learning as in Fujimoto et al. (2018), we compute

$$Q_{\boldsymbol{w}}\left(\boldsymbol{s}, \boldsymbol{\mu}_{\boldsymbol{\theta}}(\boldsymbol{s}, \boldsymbol{z})\right) = \min_{j=1,2} Q_{\boldsymbol{w}_j}\left(\boldsymbol{s}, \boldsymbol{\mu}_{\boldsymbol{\theta}}(\boldsymbol{s}, \boldsymbol{z})\right) \tag{10}$$

in our implementation. The second term in Equation 8 is approximated as the mean squared error, similar to that in the standard implementation of VAE. Based on Equation 7 and Equation 8, we obtain the objective function for training the mixture policy as follows:

$$\mathcal{L}_{\text{ML}}(\boldsymbol{\theta}, \boldsymbol{\phi}) = \sum_{i=1}^{N} f^{\pi}(\boldsymbol{s}_i, \boldsymbol{a}_i) \ell_{\text{cvae}}(\boldsymbol{s}_i, \boldsymbol{a}_i; \boldsymbol{\theta}, \boldsymbol{\phi}). \tag{11}$$

This objective can be regarded as the weighted maximum likelihood (Kober & Peters, 2011) of a mixture policy. Our objective function can be viewed as reconstructing the state-action pairs with adaptive weights, similar to that in Peters & Schaal (2007); Nair et al. (2020). Therefore, the policy samples actions within the support and does not evaluate out-of-distribution actions. The primary difference between the proposed and existing methods (Peters & Schaal, 2007; Nair et al., 2020) is that: the use of a mixture of policies conditioned on discrete latent variables in our approach can be regarded as dividing the state-action space. For example, in AWAC (Nair et al., 2020), a unimodal policy was used to reconstruct all of the "good" actions in the given dataset. However, in the context of offline RL, the given dataset may contain samples collected by diverse behaviors and enforcing the policy to cover all modes in the dataset can degrade the resulting performance. In our approach, policy $\pi_{\boldsymbol{\theta}}(\boldsymbol{a}|\boldsymbol{s}, \boldsymbol{z})$ is encouraged to mimic the state-action pairs that are assigned to the same values of $\boldsymbol{z}$ without mimicking the actions that are assigned to different values of $\boldsymbol{z}$.

**Approximation gap** When training a stochastic policy, the first term in Equation 7 can be directly maximized because it is trivial to compute the expected log-likelihood $\mathbb{E}[\log \pi(a|s)]$. However, when a policy is given by a mixture of deterministic policies, it is not trivial. For this reason, we used the variational lower bound in Equation 8. In addition, $\mathbb{E}[\log \pi(a|s, z)]$ is replaced with MSE as in VAE. As described in Cremer et al. (2018), the use of the objective in Equation 8 instead of $\log \pi(a|s)$ leads to the approximation gap, $\mathbb{E}[\log \pi(a|s)] - \ell_{\text{cvae}}(s, a)$ as in VAE. Although addressing the approximation gap using techniques investigated in Cremer et al. (2018) may improve the performance of DMPO, such investigation is left for future work.

## 4.2 Mutual-information-based regularization

To improve the performance of the mixture of deterministic policies, we propose a regularization technique for a mixture policy based on the mutual information (MI) between $\boldsymbol{z}$ and the state action pair $(\boldsymbol{s}, \boldsymbol{a})$, which we denote by $I(\boldsymbol{z}; \boldsymbol{s}, \boldsymbol{a})$. As shown in Barber & Agakov (2003), the variational lower bound of $I(\boldsymbol{z}; \boldsymbol{s}, \boldsymbol{a})$ is given as follows:

$$\begin{aligned} I(\boldsymbol{s}, \boldsymbol{a}; \boldsymbol{z}) = H(\boldsymbol{z}) - H(\boldsymbol{z}|\boldsymbol{s}, \boldsymbol{a}) &= \mathbb{E}_{(\boldsymbol{s}, \boldsymbol{a}, \boldsymbol{z}) \sim p_\pi}\left[\log p(\boldsymbol{z}|\boldsymbol{s}, \boldsymbol{a})\right] + H(\boldsymbol{z}) \\ &= \mathbb{E}_{(\boldsymbol{s}, \boldsymbol{a}) \sim \beta(\boldsymbol{s}, \boldsymbol{a})}\left[D_{\text{KL}}\left(p(\boldsymbol{z}|\boldsymbol{s}, \boldsymbol{a}) || g_{\boldsymbol{\psi}}(\boldsymbol{z}|\boldsymbol{s}, \boldsymbol{a})\right)\right] + \mathbb{E}_{(\boldsymbol{s}, \boldsymbol{a}, \boldsymbol{z}) \sim p}\left[\log g_{\boldsymbol{\psi}}(\boldsymbol{z}|\boldsymbol{s}, \boldsymbol{a})\right] + H(\boldsymbol{z}) \\ &\geq \mathbb{E}_{(\boldsymbol{s}, \boldsymbol{a}, \boldsymbol{z}) \sim p}\left[\log g_{\boldsymbol{\psi}}(\boldsymbol{z}|\boldsymbol{s}, \boldsymbol{a})\right] + H(\boldsymbol{z}), \end{aligned} \tag{12}$$

where $g_{\boldsymbol{\psi}}(\boldsymbol{z}|\boldsymbol{s}, \boldsymbol{a})$ is an auxiliary distribution to approximate the posterior distribution $p(\boldsymbol{z}|\boldsymbol{s}, \boldsymbol{a})$.

Thus, the final objective function is as follows:

$$\mathcal{L}(\boldsymbol{\theta}, \boldsymbol{\phi}, \boldsymbol{\psi}) = \mathcal{L}_{\mathrm{ML}}(\boldsymbol{\theta}, \boldsymbol{\phi}) + \lambda \sum_{i=1}^{N} \mathbb{E}_{\boldsymbol{z} \sim p(\boldsymbol{z})} \log g_{\boldsymbol{\psi}}(\boldsymbol{z}|\boldsymbol{s}_i, \boldsymbol{\mu}_{\boldsymbol{\theta}}(\boldsymbol{s}_i, \boldsymbol{z})). \tag{13}$$

MI-based regularization using the second term in Equation 13 encourages the diversity of the behaviors encoded in the sub-policy $\pi(\boldsymbol{a}|\boldsymbol{s}, \boldsymbol{z})$. In Section 7, we empirically show that this regularization improves the performance of the proposed method.

To implement MI-based regularization, we introduced a network to represent $g_{\boldsymbol{\psi}}(\boldsymbol{z}|\boldsymbol{s}, \boldsymbol{a})$ in addition to a network that represents the posterior distribution $q_{\boldsymbol{\phi}}(\boldsymbol{z}|\boldsymbol{s}, \boldsymbol{a})$. While maximizing the objective $\mathcal{L}_{\mathrm{ML}}$ in Equation 11, both the actor $\mu_{\boldsymbol{\theta}}(s, z)$ and posterior distribution $q_{\boldsymbol{\phi}}(\boldsymbol{z}|\boldsymbol{s}, \boldsymbol{a})$ are updated; however, the auxiliary distribution $g_{\boldsymbol{\psi}}(\boldsymbol{z}|\boldsymbol{s}, \boldsymbol{a})$ is frozen. While maximizing $\sum_{i=1}^{N} \mathbb{E}_{\boldsymbol{z} \sim p(\boldsymbol{z})} \log g_{\boldsymbol{\psi}}(\boldsymbol{z}|\boldsymbol{s}_i, \boldsymbol{\mu}_{\boldsymbol{\theta}}(\boldsymbol{s}_i, \boldsymbol{z}))$, both actor $\mu_{\boldsymbol{\theta}}(s, z)$ and auxiliary distribution $g_{\boldsymbol{\psi}}(\boldsymbol{z}|\boldsymbol{s}, \boldsymbol{a})$ are updated, but the posterior distribution $q_{\boldsymbol{\phi}}(\boldsymbol{z}|\boldsymbol{s}, \boldsymbol{a})$is frozen. To maximize $\log g_{\boldsymbol{\psi}}(\boldsymbol{z}|\boldsymbol{s}_i, \boldsymbol{\mu}_{\boldsymbol{\theta}}(\boldsymbol{s}_i, \boldsymbol{z}))$, the latent variable is sampled from the prior distribution, that is, the uniform distribution in this case, and the maximization of $\log g_{\boldsymbol{\psi}}(\boldsymbol{z}|\boldsymbol{s}_i, \boldsymbol{\mu}_{\boldsymbol{\theta}}(\boldsymbol{s}_i, \boldsymbol{z}))$ is approximated by minimizing the cross entropy loss between $\boldsymbol{z}$ and $\hat{\boldsymbol{z}}$, where $\hat{\boldsymbol{z}}$ is the output of the network that represents $g_{\boldsymbol{\psi}}(\boldsymbol{z}|\boldsymbol{s}, \boldsymbol{a})$.

## 5 Training the critic for a mixture of deterministic policies

To derive the objective function for training the critic for a mixture of deterministic policies using the gating policy in Equation 3, we consider the following operator:

$$\mathcal{T}_{\boldsymbol{z}} Q_{\boldsymbol{z}} = r(\boldsymbol{s}, \boldsymbol{a}) + \gamma \mathbb{E}_{\boldsymbol{s}'} \left[ \max_{\boldsymbol{z}'} Q_{\boldsymbol{z}}(\boldsymbol{s}', \boldsymbol{\mu}(\boldsymbol{s}', \boldsymbol{z}')) \right]. \tag{14}$$

We refer to operator $\mathcal{T}_{\boldsymbol{z}}$ as the *latent-max-Q operator*. Following the method in Ghasemipour et al. (2021), we prove the following theorems.

**Theorem 5.1.** *In the tabular setting, $\mathcal{T}_{\boldsymbol{z}}$ is a contraction operator in the $\mathcal{L}_{\infty}$ norm. Hence, with repeated applications of $\mathcal{T}_{\boldsymbol{z}}$, any initial Q function converges to a unique fixed point.*

The proof of Theorem 5.1 is provided in Appendix C.

**Theorem 5.2.** *Let $Q_{\boldsymbol{z}}$ denote the unique fixed point achieved in Theorem 5.1 and $\pi_{\boldsymbol{z}}$ denote the policy that chooses the latent variable as $\boldsymbol{z} = \arg\max_{\boldsymbol{z}'} Q(\boldsymbol{s}, \boldsymbol{\mu}(\boldsymbol{s}, \boldsymbol{z}'))$ and outputs the action given by $\boldsymbol{\mu}(\boldsymbol{s}, \boldsymbol{z})$ in a deterministic manner. Then $Q_{\boldsymbol{z}}$ is the Q-value function corresponding to $\pi_{\boldsymbol{z}}$.*

*Proof.* (Theorem 5.2) Rearranging Equation 14 with $\boldsymbol{z}' = \arg\max Q_{\boldsymbol{z}}(\boldsymbol{s}', \boldsymbol{\mu}(\boldsymbol{s}', \boldsymbol{z}'))$, we obtain

$$\mathcal{T}_{\boldsymbol{z}} Q_{\boldsymbol{z}} = r(\boldsymbol{s}, \boldsymbol{a}) + \gamma \mathbb{E}_{\boldsymbol{s}'} \mathbb{E}_{\boldsymbol{a}' \sim \pi_{\boldsymbol{z}}} \left[ Q_{\boldsymbol{z}}(\boldsymbol{s}', \boldsymbol{a}') \right]. \tag{15}$$

Because $Q_{\boldsymbol{z}}$ is the unique fixed point of $\mathcal{T}_{\boldsymbol{z}}$, we have our result. □

These theorems reveal that the latent-max-Q operator, $\mathcal{T}_{\boldsymbol{z}}$, retains the contraction and fixed-point existence properties. Based on these results, we estimate the Q-function by applying the latent-max-Q operator. In our implementation, we employed double-clipped Q-learning (Fujimoto et al., 2018). Thus, given dataset $\mathcal{D}$, the critic is trained by minimizing

$$\mathcal{L}(\boldsymbol{w}_j) = \sum_{(\boldsymbol{s}_i, \boldsymbol{a}_i, \boldsymbol{s}'_i, r_i) \in \mathcal{D}} \left\| Q_{\boldsymbol{w}_j}(\boldsymbol{s}_i, \boldsymbol{a}_i) - y_i \right\|^2 \tag{16}$$

for $j = 1, 2$, where target value $y_i$ is computed as

$$y_i = r_i + \gamma \max_{\boldsymbol{z}' \in \mathcal{Z}} \min_{j=1,2} Q_{\boldsymbol{w}'_j}(\boldsymbol{s}'_i, \boldsymbol{\mu}_{\boldsymbol{\theta}'}(\boldsymbol{s}'_i, \boldsymbol{z}')). \tag{17}$$

---

**Algorithm 1** Deterministic mixture policy optimization (DMPO)

---

Initialize the actor $\boldsymbol{\mu_\theta}$, critic $Q_{\boldsymbol{w}_j}$ for $j = 1, 2$, and the posterior $q_\phi(\boldsymbol{z}|\boldsymbol{s}, \boldsymbol{a})$
**for** $t = 1$ **to** $T$ **do**
    Sample a minibatch $\{(\boldsymbol{s}_i, \boldsymbol{a}_i, \boldsymbol{s}'_i, r_i)\}_{i=1}^M$ from $\mathcal{D}$
    **for** each element $(\boldsymbol{s}_i, \boldsymbol{a}_i, \boldsymbol{s}'_i, r_i)$ **do**
        Compute the target value as
        $y_i = r + \gamma \max_{\boldsymbol{z}' \in \mathcal{Z}} \min_{j=1,2} Q_{\boldsymbol{w}'_j}(\boldsymbol{s}'_i, \boldsymbol{\mu_{\theta'}}(\boldsymbol{s}'_i, \boldsymbol{z}'))$
    **end for**
    Update the critic by minimizing $\sum_{i=1}^M \left\| y_i - Q_{\boldsymbol{w}_j}(\boldsymbol{s}_i, \boldsymbol{a}_i) \right\|^2$ for $j = 1, 2$
    **if** $t \mod d_{\text{interval}} = 0$ **then**
        Update the actor and the posterior by maximizing Equation 11
        (optionally) Update the actor by maximizing $\sum_{i=1}^M \mathbb{E}_{\boldsymbol{z} \sim p(\boldsymbol{z})} \log g_\psi(\boldsymbol{z}|\boldsymbol{s}_i, \boldsymbol{\mu_\theta}(\boldsymbol{s}_i, \boldsymbol{z}))$
    **end if**
**end for**

---

## 6 Practical implementation

The proposed DMPO algorithm is summarized as Algorithm 1. Similar to that in TD3 (Fujimoto et al., 2018), the actor is updated once after $d_{\text{interval}}$ updates of the critics. In our implementation, we set $d_{\text{interval}} = 2$. The discrete latent variable is represented by a one-hot vector, and we used the Gumbel-Softmax trick to sample the discrete latent variable in a differentiable manner (Jang et al., 2017; Maddison et al., 2017). Herein, following Jang et al. (2017); Maddison et al. (2017), we assume that $\boldsymbol{z}$ is a categorical variable with class probabilities $\alpha_1, \alpha_2, \ldots, \alpha_k$ and categorical samples are encoded as $k$-dimensional one-hot vectors lying on the corners of the $(k-1)$-dimensional simplex, $\Delta^{k-1}$. In the Gumbel-Softmax trick, sample vectors $\tilde{\boldsymbol{z}} \in \Delta^{k-1}$ are generated as follows:

$$\tilde{z}_i = \frac{\exp\left((\log \alpha_i + G_i)/\lambda\right)}{\sum_{j=1}^k \exp\left((\log \alpha_j + G_j)/\lambda\right)}, \tag{18}$$

where $G_i$ is sampled from the Gumbel distribution as $G_i \sim \text{Gumbel}(0, 1)$, and $\lambda$ is the temperature. As the temperature $\lambda$ approaches 0, the distribution of $\tilde{\boldsymbol{z}}$ smoothly approaches the categorical distribution $p(\boldsymbol{z})$. As in prior work on the VAE with the Gumbel-Softmax trick (Dupont, 2018), we set $\lambda = 0.67$ in our implementation of DMPO.

There are several promising ways for learning discrete latent variable (van den Oord et al., 2017; Razavi et al., 2019), and investigating the best way of learning the discrete latent variable is left for future work.

Additionally, we employed the state normalization method used in TD3+BC (Fujimoto & Gu, 2021). During preliminary experiments, we found that when $f^\pi(\boldsymbol{s}, \boldsymbol{a}) = \exp(bA^\pi(\boldsymbol{s}, \boldsymbol{a}))$ in Equation 11, scaling factor $b$ has non-trivial effects on performance and the best value of $b$ differs for each task. To avoid changing the scaling parameter for each task, we used the normalization of the advantage function as

$$f^\pi(\boldsymbol{s}, \boldsymbol{a}) = \exp\left(\frac{\alpha\left(A^\pi(\boldsymbol{s}, \boldsymbol{a}) - \max_{(\tilde{\boldsymbol{s}}, \tilde{\boldsymbol{a}}) \in \mathcal{D}_{\text{batch}}} A^\pi(\tilde{\boldsymbol{s}}, \tilde{\boldsymbol{a}})\right)}{\max_{(\tilde{\boldsymbol{s}}, \tilde{\boldsymbol{a}}) \in \mathcal{D}_{\text{batch}}} A^\pi(\tilde{\boldsymbol{s}}, \tilde{\boldsymbol{a}}) - \min_{(\tilde{\boldsymbol{s}}, \tilde{\boldsymbol{a}}) \in \mathcal{D}_{\text{batch}}} A^\pi(\tilde{\boldsymbol{s}}, \tilde{\boldsymbol{a}})}\right), \tag{19}$$

where $\mathcal{D}_{\text{batch}}$ is a mini-batch sampled from the given dataset $\mathcal{D}$ and $\alpha$ is a constant. We set $\alpha = 10$ for mujoco tasks and $\alpha = 5.0$ for antmaze tasks in our experiments. For other hyperparameter details, please refer to the Appendix F.

## 7 Experiments

We investigated the effect of policy structure on the resulting performance and training errors of critics. In the first experiment, we performed a comparative assessment of TD3+BC (Fujimoto & Gu, 2021), AWAC (Nair et al., 2020) and DMPO on a toy problem where the distribution of samples in a given dataset is multimodal.

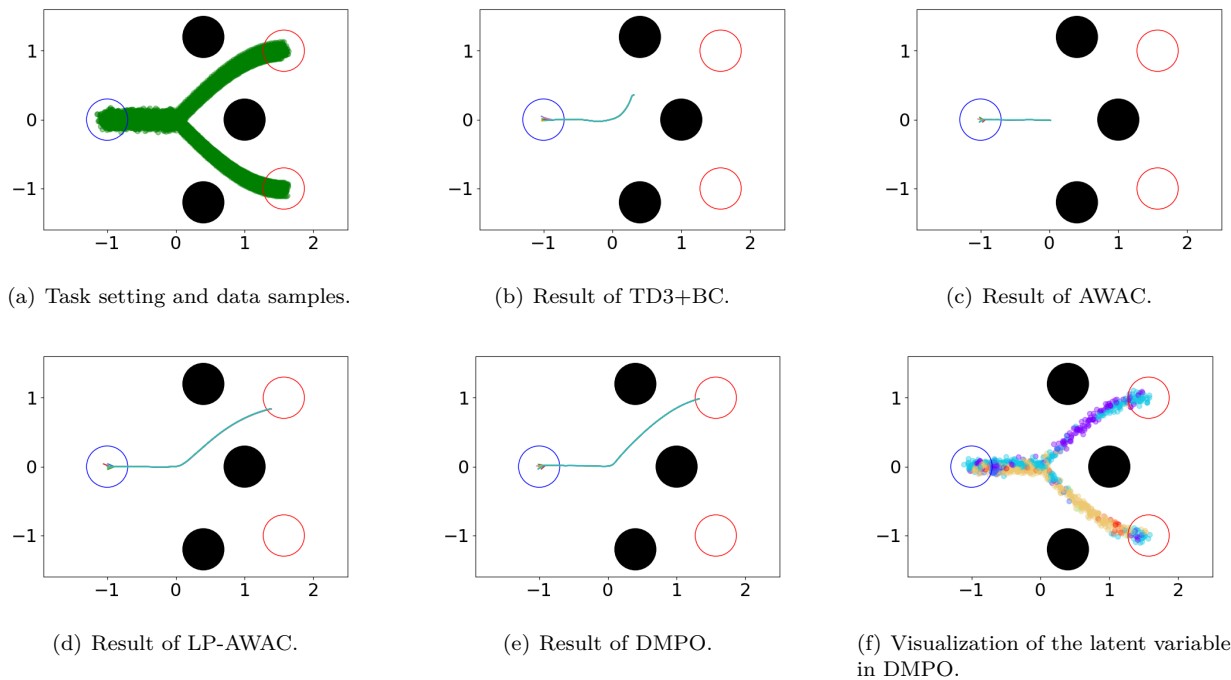

Figure 2: Performance on a simple task with multimodal data distribution.

Table 1: Algorithm setup in the experiment.

|  | TD3+BC | AWAC | LP-AWAC | DMPO |
|---|---|---|---|---|
| Critic training | double-clipped Q-learning | double-clipped Q-learning | double-clipped Q-learning | double-clipped Q-learning |
| Policy type | monolithic & deterministic | monolithic & stochastic | deterministic on continuous latent action space | mixture & deterministic |
| Regularization | BC term | none | none | none |
| State | normalized | normalized | normalized | normalized |
| Advantage normalization | - | yes | yes | yes |

Further, we conducted a quantitative comparison between the proposed and baseline methods with D4RL benchmark tasks (Fu et al., 2020). In the following section, we refer to the proposed method based on the objective in Equation 11 as DMPO, and a variant of the proposed method with MI-based regularization in Equation 13 as infoDMPO. In both, the toy problem and the D4RL tasks, we used the author-provided implementations of TD3+BC, and our implementations of DMPO and AWAC are based on the author-provided implementation of TD3+BC. Our implementation is available at `https://github.com/TakaOsa/DMPO`.

## 7.1 Multimodal data distribution on toy task

To show the effect of multimodal data distribution in a given dataset, we evaluated the performance of TD3+BC, AWAC, and DMPO on a toy task, as shown in Figure 2. We also evaluated the variant of AWAC that employs the policy structure used in LAPO Chen et al. (2022), which we refer to as LP-AWAC. In LP-AWAC, the continuous latent representations of state action pairs are learned using conditional VAE with advantage weighting, and a deterministic policy that outputs actions in the learned latent space is trained using DDPG. We found that the authors' implementation of LAPO[1] includes techniques to improve

---

[1] `https://github.com/pcchenxi/LAPO-offlienRL`

performance, such as action normalization and clipping of the target value for the state-value function. While LP-AWAC employs the policy structure proposed by Chen et al. (2022), the implementation of LP-AWAC is largely modified from that of the authors' implementation of LAPO to minimize the difference among our implementation of AWAC, mixAWAC, LP-AWAC, and DMPO. LP-AWAC can be considered as a baseline method that incorporates a continuous latent variable in its policy structure. The implementation details of LP-AWAC is described in Appendix F. The differences between the compared methods are summarized in Table 1. In our implementation of AWAC, LP-AWAC and DMPO, we used state normalization and double-clipped Q-learning as in TD3+BC and the normalization of the advantage function described in Section 6. The difference among AWAC, LP-AWAC and DMPO indicates the effect of the policy representation.

In this toy task, the agent is represented as a point mass, the state is the position of the point mass in two-dimensional space, and the action is the small displacement of the point mass. There are two goals in this task, which are indicated by red circles in Figure 2. The blue circle denotes the starting position in Figure 2, and there are three obstacles, which are indicated by solid black circles. In this task, the reward is sparse: when the agent reaches one of the goals, it receives a reward of 1.0, and the episode ends. If the agent makes contact with one of the obstacles, the agent receives a reward of -1.0, and the episode ends. In the given dataset, trajectories for the two goals are provided, and there is no information on which goal the agent is heading to.

The scores are summarized in Table 2. Among the evaluated methods, only DMPO successfully solved this task. The policy obtained by TD3+BC did not reach its goal in a stable manner, as shown in Figure 2(b). Similarly,

Table 2: Performance on the toy task.

| TD3+BC | AWAC | LP-AWAC | DMPO |
|---|---|---|---|
| -0.2± 0.4 | 0.33± 0.44 | 0.7±0.6 | **1.0±0.0** |

as shown in Figure 2(c), the policy learned by AWAC often slows down around point $(0, 0)$ and fails to reach the goal. This behavior implies that AWAC attempts to average over multiple modes of the distribution. In contrast, the policy learned by DMPO successfully reaches one of the goals. Because the main difference between AWAC and DMPO is the policy architecture, the result shows that the unimodal policy distribution fails to deal with the multimodal data distribution, whereas the mixture policy employed in DMPO successfully dealt with it. Similarly, the performance of LP-AWAC is significantly better than TD3+BC and AWAC, demonstrating the benefit of the policy structure based on the latent action space. On the other hand, the performance of DMPO was better than that of LP-AWAC, indicating the advantage of using the discrete latent variable in offline RL. The activation of the sub-policies is visualized in Figure 2(e). The color indicates the value of the discrete latent variable given by the gating policy, $z^* = \arg\max_{\boldsymbol{z}} Q_{\boldsymbol{w}}(\boldsymbol{s}, \boldsymbol{\mu}(\boldsymbol{s}, \boldsymbol{z}))$. Figure 2(d) shows that different sub-policies are activated for different regions, thereby indicating that DMPO appropriately divides the state-action space.

## 7.2 Effect of policy structure

We investigated the effect of policy structure by comparing the proposed method with existing methods that incorporate the importance weight based on the advantage function. We used AWAC as baseline methods. To investigate the difference between the mixture of the stochastic policies and the mixture of the deterministic policies, we evaluated a variant of AWAC with Gaussian mixture policies, which we refer to as mixAWAC. For mixAWAC, the Gumbel-Softmax trick was used to sample the discrete latent variable. All baseline methods used double-clipped Q-learning for the critic in this experiment.

The implementations of AWAC and DMPO were identical to those used in the previous experiment. In our evaluation, $|Z| = 8$ was used for DMPO and infoDMPO. Appendix D presents the effect of the dimensionality of the discrete latent variables. In this study, we evaluated the baseline methods with mujoco-v2 and antmaze-v0 tasks.

### 7.2.1 Performance score on D4RL

A comparison between AWAC, mixAWAC, LP-AWAC, and DMPO is presented in Table 3. These methods incorporate importance weights based on the advantage function with different policy structures. Therefore, the differences between these methods indicate the effect of policy structure. In our experiments, we did not observe significant differences in the performance of AWAC and mixAWAC. This result indicates that

Table 3: Comparison with methods incorporating advantage-weighting using D4RL-v2 datasets. Average normalized scores over the last 10 test episodes and five seeds are shown. The boldface text indicates the best performance. HC = HalfCheetah, HP = Hopper, WK = Walker2d.

| | | AWAC | mixAWAC | LP-AWAC | DMPO |
|---|---|---|---|---|---|
| Expert | HC | 94.8±0.2 | 94.0±0.5 | 93.7±0.4 | **97.0±1.0** |
| | HP | 109.8±2.9 | **111.8±0.8** | 104.3±5.5 | 93.6±15.1 |
| | WK | 111.0±0.2 | 110.5±0.3 | 110.7±0.1 | **111.4±0.3** |
| Med.-E | HC | 92.7±0.8 | 92.1±0.6 | 92.5±0.4 | 91.1±3.4 |
| | HP | **98.6±10.7** | **102.0±17.5** | 90.5±21.6 | 78.4±19.0 |
| | WK | 109.2±0.3 | 109.1±0.4 | 109.1±0.4 | **109.9±0.4** |
| Med.-R | HC | 40.9±0.6 | 41.5±0.4 | 39.8±0.3 | **45.2±0.8** |
| | HP | 38.2±9.4 | 41.2±4.7 | 46.1±8.1 | **89.2±8.1** |
| | WK | 65.0±15.7 | 67.7±8.8 | 50.2±5.5 | **82.1±3.8** |
| Med. | HC | 44.3±0.2 | 45.1±0.3 | 44.0±0.4 | **47.5±0.4** |
| | HP | 57.5±3.0 | 57.2±3.9 | 52.8±3.8 | **71.2±6.5** |
| | WK | **81.0±2.5** | 78.7 ± 4.8 | 77.4±2.7 | 79.4±4.7 |
| Rand. | HC | 3.2±1.3 | 2.2±0.0 | 4.1±2.3 | **15.8±1.6** |
| | HP | 7.3±0.9 | 8.2±0.2 | 8.4±0.6 | **12.0±10.0** |
| | WK | 3.1±1.0 | **4.9±1.1** | **4.0±1.2** | 2.5±2.6 |
| Antmaze | umaze | 49.8±6.2 | 57.4±6.2 | 56.6±4.1 | **83.6±4.5** |
| | umaze-d. | 53.8±13.0 | 46.8±6.9 | **66.6±5.5** | 43.2±7.8 |
| | med.-p. | 0.0±0.0 | 0.0±0.0 | 0.0±0.0 | **77.0±5.1** |
| | med.-d. | 0.0±0.0 | 0.0±0.0 | 0.0±0.0 | **56.8±27.2** |
| | large-p. | 0.0±0.0 | 0.0±0.0 | 0.0±0.0 | **1.0±1.3** |
| | large-d. | 0.0±0.0 | 0.0±0.0 | 0.0±0.0 | **4.8±9.6** |

the use of Gaussian mixture policies does not lead to performance improvement. However, the performance of DMPO matched or exceeded that of AWAC, except for the Hopper-expert and Hopper-medium-expert tasks. This result also confirms that the use of a mixture of deterministic policies is beneficial for these tasks, although the benefits would be task-dependent.

The difference between mixAWAC and DMPO implies a difference between a Gaussian mixture policy and a mixture of deterministic policy. In a Gaussian mixture policy, there is a possibility that one of the Gaussian components covers a large action space and interpolates the separate modes of action. If this happens, out-of-distribution actions will be generated by the learned policy. However, in a mixture of the deterministic policy, there is no such possibility that one of the components covers a large action space.

In addition, DMPO outperformed LP-AWAC on mujoco-v2 and antmaze-v0 tasks. As the difference between DMPO and LP-AWAC indicates the difference between the discrete and continuous latent representations in our framework, this result also indicates that the use of a discrete latent variable is beneficial for offline RL tasks. A comparison with additional baseline methods is provided in Appendix E.

### 7.2.2 Critic loss function

To investigate the effect of the policy structure on the critic loss function, we compared the value of the critic loss function among AWAC, mixAWAC, LP-AWAC, and DMPO. The normalized scores and value of the critic loss function during training are depicted in Figure 3. The value of the critic loss given by Equation 16 is plotted for every 5,000 updates. Previous studies have indicated that the critic loss value can accumulate over iterations (Brandfonbrener et al., 2021). Figure 3 shows the accumulation of the critic loss in AWAC on mujoco-v2 tasks. The difference between AWAC and mixAWAC indicates that using a Gaussian mixture policy often reduces the accumulation of the critic loss. The critic loss of mixAWAC is lower than that of AWAC in halfcheetah-medium-replay-v2 and halfcheetah-medium-expert-v2 tasks. This result shows that the use of a multimodal policy can reduce the accumulation of the critic loss in offline RL. In addition, the critic loss of DMPO is even lower than that of mixAWAC, and the result demonstrates that using a mixture of deterministic policies can further reduce the critic loss than using a Gaussian mixture policy. These results indicate that using a mixture of deterministic policies can reduce the generation of out-of-distribution actions, which is essential for offline RL.

Regarding LP-AWAC, the critic loss value increased rapidly at the beginning of the training. Although the critic loss value often decreases at the end of the LP-AWAC training, the critic loss value is still higher than

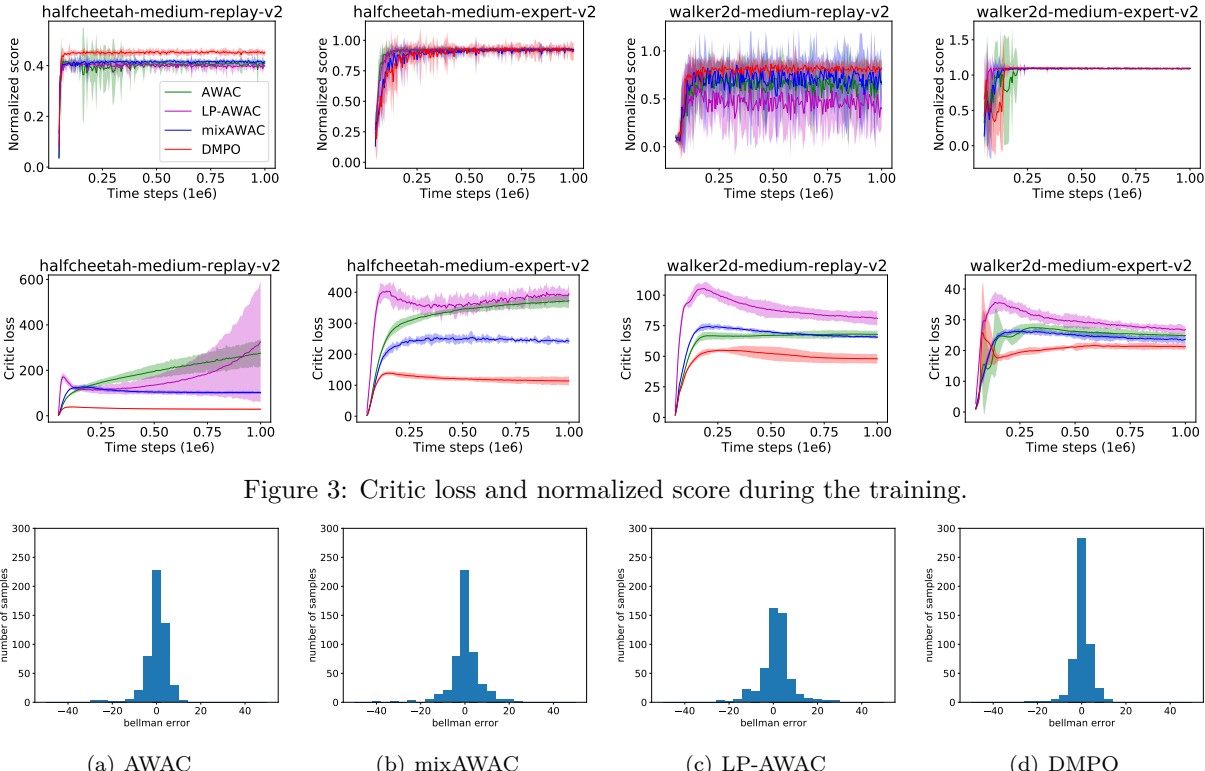

Figure 3: Critic loss and normalized score during the training.

(a) AWAC      (b) mixAWAC      (c) LP-AWAC      (d) DMPO

Figure 4: Histogram of the Bellman errors after 20k steps on the halfcheetah-medium-replay task.

that of DMPO. The surge in the critic loss value indicates the generation of out-of-distribution actions during training in LP-AWAC. Importantly, in DMPO, the value of the critic loss is clearly lower, and the performance of the policy is better than that of LP-AWAC. This result indicates that the use of a discrete latent variable can be more effective than using a continuous latent variable on these tasks. In Brandfonbrener et al. (2021), it was shown that the accumulation of critic loss values can be reduced by introducing regularization. Our results indicate that the use of a mixture policy can also mitigate the accumulation of critic loss in offline RL, which suggests the importance of incorporating inductive bias in the policy structure. However, it is worth noting that the reduction in the critic loss given by Equation 16 does not necessarily improve the policy performance. In halfcheetah-medium-expert-v2, although the critic loss was significantly lower in DMPO than in AWAC, there was no significant difference in performance between DMPO and AWAC. Recently, Fujimoto et al. indicated that a lower value of the critic loss given by Equation 16 does not necessarily mean better performance, and the observation in Fujimoto et al. (2022) aligns with our experiments. The metric to measure the accuracy of the value estimation is still an open problem in RL.

As another qualitative result, Figure 4 shows the histograms of the Bellman error after training with 20 thousand steps. The Bellman error in mixAWAC is distributed more widely than that in AWAC, indicating that the use of the Gaussian mixture policy can increase the variance during the training of the critic. In contrast, the distribution of the Bellman error in DMPO is more narrow than those of AWAC, mixAWAC, and LP-AWAC, indicating that the use of the mixture of deterministic policies may lead to reducing the variance during the critic training. This variance reduction could also be considered a reason why DMPO outperformed baseline methods.

## 7.3 Comparison with prevalent baselines

We compared the performance of the proposed method with that of prevalent baselines. As baseline methods, we used TD3+BC, CQL (Kumar et al., 2020), IQL (Kostrikov et al., 2022), LAPO, and Diffusion QL(Wang et al., 2023). CQL incorporates conservative critic update and the entropy regularization. In the experiments reported in this section, we used the authors' implementation of LAPO. Diffusion QL is recently proposed by

Table 4: Results on mujoco tasks using D4RL-v2 datasets and AntMaze tasks. Average normalized scores over the last 10 test episodes and five seeds are shown. HC = HalfCheetah, HP = Hopper, WK = Walker2d. "Diff. QL" represents Diffusion QL proposed in Wang et al. (2023).

| | | TD3+BC (re-run) | CQL (re-run) | IQL (re-run) | LAPO (re-run) | Diff. QL (re-run) | DMPO (ours) | infoDMPO (ours) |
|---|---|---|---|---|---|---|---|---|
| Expert | HC | **96.3±0.9** | 22.0±9.6 | **96.1±1.5** | 95.4±0.3 | 86.3±15.9 | **97.0±1.0** | 95.6±2.0 |
| | HP | **109.9±2.5** | 105.8±3.8 | 98.4±13.1 | **110.9±2.3** | 84.3±24.2 | 93.6±15.1 | 107.5±2.9 |
| | WK | 110.2±0.4 | 108.9±0.4 | **112.6±0.3** | 111.5±0.2 | 109.0±0.6 | 111.4±0.3 | **112.1±0.4** |
| Med.-E | HC | 89.4±7.2 | 38.4±8.4 | 90.7±4.3 | 94.3±1.1 | 83.8±15.3 | **91.1±3.4** | 91.4±2.5 |
| | HP | 95.5±9.4 | 88.4±15.9 | 73.9±32.6 | **110.5±1.2** | 88.1±25.7 | 78.4±19.0 | 94.5±14.9 |
| | WK | 110.2±0.3 | 109.2±1.9 | **111.4±1.1** | 111.0±0.2 | 110.1±0.6 | 109.9±0.4 | 110.1±0.7 |
| Med.-R | HC | 44.7±0.4 | **46.9±0.3** | 43.6±1.4 | 41.9±1.0 | 45.6±0.6 | 45.2±0.8 | **46.7±0.6** |
| | HP | 73.8±18.9 | 95.5±1.7 | 90.6±14.3 | 59.7±14.2 | 56.1±24.0 | 89.2±8.1 | **98.5±2.0** |
| | WK | 64.5±17.0 | 77.5±3.1 | 82.2±3.6 | 50.3±18.6 | 84.1±17.0 | 82.1±3.8 | **86.7±3.2** |
| Med. | HC | **48.2±0.3** | **48.2±0.4** | **48.2±0.2** | 45.7±0.3 | 46.7±0.7 | 47.5±0.4 | **48.6±0.4** |
| | HP | 61.0±4.2 | 77.4±4.0 | 61.2±3.5 | 56.2±5.1 | 57.1±11.4 | 71.2± 6.5 | **86.4±7.6** |
| | WK | **84.7±1.3** | 81.5±2.5 | 82.9±6.0 | 80.5±1.8 | 62.1±20.6 | 79.4±4.7 | **85.0±0.8** |
| Rand. | HC | 11.5±0.6 | 24.1±1.5 | 12.6±4.6 | **27.1±1.0** | 17.5±0.2 | 15.8±1.6 | 16.3±1.2 |
| | HP | 8.7±0.3 | 2.2±1.9 | 7.4±0.3 | 15.2±8.6 | 7.8±0.5 | 12.0±10.0 | **20.4±9.8** |
| | WK | 1.4±1.9 | **4.3±7.9** | **5.5±1.6** | 2.2±1.5 | **6.2±3.4** | 2.5± 2.6 | 2.3±2.0 |
| | Total | 1010.0 | 930.3 | 1017.3 | 1012.4 | 942.1 | 1026.5 | **1102.1** |

Wang et al. (2023) and employs a diffusion model as a policy. We used the author implementation of Diffusion QL, and the results of Diffusion QL are based on the offline model selection reported in Wang et al. (2023). IQL employs expectile regression for learning the critic to address the issue of generating out-of-distribution actions during training. Because the aim of our study is to investigate the policy structure, the approach of IQL, which address the critic learning, is orthogonal to ours. IQL is the state-of-the-art method for antmaze task on D4RL, which involves dealing with long horizons and requires "stitching" together sub-trajectories in a given dataset (Fu et al., 2020). In the implementation of IQL, several techniques, such as scheduling of the learning rate, were used to improve the performance. To compete with IQL on antmaze task, we also used techniques used in Chen et al. (2022). Therefore, the implementations of DMPO and infoDMPO for antmaze tasks are slightly different from those for other tasks. In our preliminary experiment, we evaluated IQL using the techniques proposed in Chen et al. (2022), and observed that the original implementation of IQL showed better performance. Therefore, we used with the original implementation of IQL for comparison. In this experiment, we used the mujoco-v2, antmaze-v0, and adroit tasks on D4RL.

A comparison of TD3+BC, CQL, and IQL is presented in Tables 4, 5, and 6. The boldface text indicates the best performance. In mujoco-v2 tasks, the performance of DMPO is comparable/superior to that of the state-of-the-art methods. In addition, infoDMPO, which employs MI-based regularization, outperformed DMPO on various tasks, and infoDMPO showed the best performance for 10 tasks among 15 mujoco-v2 tasks. This result shows that encouraging the diversity of sub-policies using the proposed MI-based regularization is effective for DMPO.

The advantages of DMPO and infoDMPO over TD3+BC and CQL are apparent for antmaze tasks. TD3+BC and CQL did not work satisfactorily on antmaze tasks, indicating that techniques used in these algorithms are not effective for such tasks. The performance of DMPO(ant ver.) and infoDMPO(ant ver.) on antmaze tasks is comparable to that of IQL and Diffusion QL, which are the state-of-the-art method for these tasks. We observed similar results for adroit tasks. DMPO and infoDMPO clearly outperformed TD3+BC and CQL, and the performance of DMPO and infoDMPO was comparable to IQL. Considering that infoDMPO outperformed IQL on mujoco-v2 task, overall performance of infoDMPO is better than that of IQL. This result reveals that the use of a mixture of deterministic policies can result in a significant performance improvement in offline RL.

We also provide the result regarding the computational cost of infoDMPO in Table 7. We used a workstation with GPU RTX A6000 and CPU Core i9-10980XE for this evaluation. The results indicate that the

Table 5: Results on AntMaze tasks. Average normalized scores over the last 100 test episodes and five seeds are shown.

| | | TD3+BC (re-run) | CQL (re-run) | IQL (re-run) | LAPO (re-run) | Diff. QL (re-run) | DMPO (ant ver.) (ours) | infoDMPO (ant ver.) (ours) |
|---|---|---|---|---|---|---|---|---|
| Antmaze | umaze | 92.8±2.7 | 73.0±4.9 | 87.4±4.5 | **97.2±2.7** | 80.4±35.3 | 92.8±2.1 | 89.4±5.1 |
| | umaze-d. | 45.0±22.2 | 43.8±4.4 | **64.6±5.6** | 57.4±11.7 | 8.0±21.9 | 32.6±25.6 | 34.8±18.0 |
| | med.-p. | 0.0±0.0 | 9.0±6.4 | **74.6±3.1** | 73.8±4.8 | 60.5±48.8 | 63.0±13.0 | 62.6±6.8 |
| | med.-d. | 0.0±0.0 | 3.8±4.2 | 73.8±7.1 | 81.0±3.6 | 12.4±30.0 | 75.0±8.5 | **82.8±4.4** |
| | large-p. | 0.0±0.0 | 0.0±0.0 | 39.0±7.2 | 27.6±13.3 | **44.4±48.5** | 42.2±23.0 | **47.4±14.5** |
| | large-d. | 0.0±0.0 | 0.0±0.4 | 48.0±9.0 | 26.2±17.5 | 48.6±48.8 | **56.6±4.5** | 38.0±4.8 |

Table 6: Results on adroit tasks using the average normalized scores over the last 10 test episodes and five seeds.

| | | TD3+BC (re-run) | CQL($\rho$) (re-run) | IQL (re-run) | LAPO (re-run) | Diff. QL (re-run) | DMPO (ours) | infoDMPO (ours) |
|---|---|---|---|---|---|---|---|---|
| Human | pen | 0.8±8.0 | **98.3±81.8** | 88.8±21.2 | 78.9±14.1 | 42.1±57.5 | 86.1± 8.8 | **94.8±16.5** |
| | hammer | 0.9±0.8 | -7.1±0.1 | 1.0±0.2 | 1.1±0.4 | 0.3±0.2 | 1.2±0.2 | **2.4±0.9** |
| | door | -0.3±0.0 | -3.3±7.8 | 2.4±2.1 | 3.2±1.6 | -0.4±0.0 | 1.3±1.5 | **4.2±3.1** |
| | relocate | -0.3±0.0 | **0.3±2.4** | 0.0±0.0 | 0.0±0.0 | -0.0±0.0 | 0.0± 0.1 | **0.1±0.0** |
| Cloned | pen | 0.5±7.0 | -1.7±1.5 | 39.2±15.4 | 25.6±12.2 | 19.0±42.2 | 36.0±17.7 | **46.4±16.7** |
| | hammer | 0.2±0.0 | -7.0±0.1 | 0.9±0.4 | 0.7±0.4 | 0.2±0.0 | 0.8±0.6 | **1.2±0.3** |
| | door | -0.3±0.0 | -9.4±0.0 | 0.6±1.2 | 0.7±0.9 | -0.3±0.0 | 0.0±0.0 | **0.8±0.8** |
| | relocate | -0.3±0.0 | -2.1±0.0 | **-0.2±0.0** | **-0.2±0.0** | **-0.2±0.1** | **-0.2±0.0** | **-0.2±0.0** |

computational cost of Diffusion QL is approximately three times higher than that of infoDMPO. Therefore, the computational cost is also an advantage of infoDMPO over Diffusion QL.

As a qualitative evaluation, we investigated the activation of sub-policies in DMPO. Activation of sub-policies in DMPO on the pen-human-v0 task is depicted in Figure 5. In Figure 5(a), the top row depicts the state at the 20th, 40th, 60th, and 80th time steps, and graphs in the middle row of the figure show the action-values of each sub-policy at each state, $Q_{\boldsymbol{w}}(\boldsymbol{s}, \boldsymbol{\mu}(\boldsymbol{s}, \boldsymbol{z}))$. Figures 5 (a) and (b) show the results for different episodes. A previous study (Smith et al., 2018) reported that in the option-critic framework Bacon et al. (2017) only a few options are activated and that the remaining options do not learn meaningful behaviors. In contrast, the results in Figure 5 show that the value of each of the sub-policies $Q_{\boldsymbol{w}}(\boldsymbol{s}, \boldsymbol{\mu}(\boldsymbol{s}, \boldsymbol{z}))$ changes over time, and various sub-policies are activated during execution. This result implies the following: meaningful sub-policies are learned in DMPO and different behaviors are adaptively used to perform complicated manipulation task.

# 8 Limitation of the proposed method

In this work, we proposed a method based on a mixture of deterministic policies, which implicitly divides the state-action space by learning discrete latent variables. While the experimental results demonstrate that our method DMPO can avoids the accumulation of the critic loss, there are problems which cannot be addressed by our approach. For example, if the dataset contains only covers subset of actions, there is the potential to overestimate the value of actions that are not contained in the dataset. In such cases, dividing the state action space is not sufficient to avoid generating OOD actions, and it will be necessary to regularization to avoid the overestimation of Q-values such as CQL Kumar et al. (2020).

In addition, while DMPO demonstrated the performance comparable to Diffusion QL on mujoco tasks in D4RL, a diffusion-model-based policy is evidently more expressive than a mixture of deterministic policies. When the data distribution in a given dataset is highly complex in offline RL, a diffusion-model-based policy should demonstrate its advantage over a mixture of deterministic policies.

Table 7: Wall clock time for training and inference.

|  | Time for training with 1 million steps | Action inference time |
|---|---|---|
| infoDMPO | 170 [min] | 1.2-1.3 [ms] |
| Diffusion QL | 600 [min] | 4.0-4.3 [ms] |

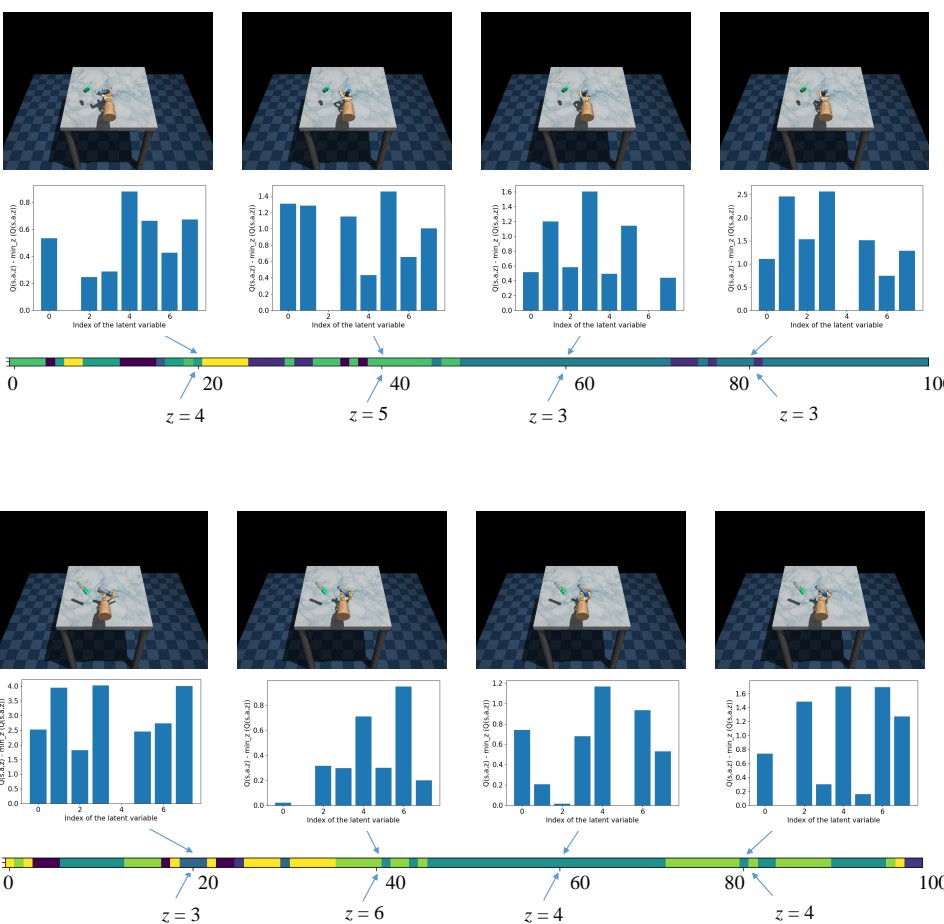

Figure 5: Visualization of sub-policy activation in the pen-human-v0 task. The top row depicts the state at the 20th, 40th, 60th, and 80th time steps; the graphs in the middle row depicts the action-values of each sub-policy at each state.

## 9    Conclusion

We presented DMPO, an algorithm for training a mixture of deterministic policies in offline RL. This algorithm can be interpreted as an approach that divides the state-action space by learning the discrete latent variable and the corresponding sub-policies in each region. In this study, we empirically investigated the effect of policy structure in offline RL. The experimental results reveal that the use of a mixture of deterministic policies can mitigate the issue of critic error accumulation in offline RL. In addition, the results indicate that the use of a mixture of deterministic policies significantly improves the performance of an offline RL algorithm. We believe that our study contributes to advancing techniques to leverage policy structure in offline RL.

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

## A  Applicability of the gating policy

In the proposed algorithm, we employ a gating policy that determines the value of the latent variable as follows:

$$z = \arg\max_{z'} Q_{\boldsymbol{w}}(s, \boldsymbol{\mu_\theta}(s, z')), \tag{20}$$

where $\boldsymbol{\mu_\theta}(s, z')$ represents the deterministic sub-policy, and $Q_{\boldsymbol{w}}(s, \boldsymbol{a})$ is the approximated Q-function. While this gating policy appears specific to the case where $Q^\pi(s, \boldsymbol{a})$ is maximized, it is applicable to other objective function such as $A^\pi(s, \boldsymbol{a})$, $\exp(Q^\pi(s, \boldsymbol{a}))$, and $\exp(A^\pi(s, \boldsymbol{a}))$. The advantage function is defined as $A^\pi(s, \boldsymbol{a}) =$

$Q^\pi(\boldsymbol{s}, \boldsymbol{a}) - V^\pi(\boldsymbol{s})$. Because the state value function $V^\pi(\boldsymbol{s})$ is independent of the action, we can obtain the following equation:

$$\arg\max_{\boldsymbol{a}} Q^\pi(\boldsymbol{s}, \boldsymbol{a}) = \arg\max_{\boldsymbol{a}} \left(Q^\pi(\boldsymbol{s}, \boldsymbol{a}) - V^\pi(\boldsymbol{s})\right) \tag{21}$$

$$= \arg\max_{\boldsymbol{a}} A^\pi(\boldsymbol{s}, \boldsymbol{a}). \tag{22}$$

Thus, we can rewrite the gating policy as

$$\boldsymbol{z} = \arg\max_{\boldsymbol{z}'} Q_{\boldsymbol{w}}(\boldsymbol{s}, \boldsymbol{\mu}_{\boldsymbol{\theta}}(\boldsymbol{s}, \boldsymbol{z}')) \tag{23}$$

$$= \arg\max_{\boldsymbol{z}'} A_{\boldsymbol{w}}(\boldsymbol{s}, \boldsymbol{\mu}_{\boldsymbol{\theta}}(\boldsymbol{s}, \boldsymbol{z}')). \tag{24}$$

Similarly, exponential function $\exp(\cdot)$ is a monotonically increasing function. Thus, the extrema of $Q^\pi(\boldsymbol{s}, \boldsymbol{a})$ is also the extrema of $\exp(Q^\pi(\boldsymbol{s}, \boldsymbol{a}))$. Consequently, we can also rewrite the gating policy as

$$\boldsymbol{z} = \arg\max_{\boldsymbol{z}'} Q_{\boldsymbol{w}}(\boldsymbol{s}, \boldsymbol{\mu}_{\boldsymbol{\theta}}(\boldsymbol{s}, \boldsymbol{z}')) \tag{25}$$

$$= \arg\max_{\boldsymbol{z}'} \exp\left(Q_{\boldsymbol{w}}(\boldsymbol{s}, \boldsymbol{\mu}_{\boldsymbol{\theta}}(\boldsymbol{s}, \boldsymbol{z}'))\right) \tag{26}$$

$$= \arg\max_{\boldsymbol{z}'} A_{\boldsymbol{w}}(\boldsymbol{s}, \boldsymbol{\mu}_{\boldsymbol{\theta}}(\boldsymbol{s}, \boldsymbol{z}')) \tag{27}$$

$$= \arg\max_{\boldsymbol{z}'} \exp\left(A_{\boldsymbol{w}}(\boldsymbol{s}, \boldsymbol{\mu}_{\boldsymbol{\theta}}(\boldsymbol{s}, \boldsymbol{z}'))\right). \tag{28}$$

Because we used this gating policy, it is deterministic in our implementation.

## B  Derivation of the variational lower bound

We employed the variational lower bound in Equation 8 to derive the objective function for the proposed method. Here, we provide a detailed derivation, which was omitted in the main manuscript. We denote the true distribution induced by the policy $\pi_{\boldsymbol{\theta}}(\boldsymbol{a}|\boldsymbol{s})$ as $p(\cdot)$, and the distribution that approximates the true distribution is denoted as $q(\cdot)$. The KL divergence between $q(\boldsymbol{x})$ and $p(\boldsymbol{x})$ is defined as

$$D_{\mathrm{KL}}\left(q(\boldsymbol{x})||p(\boldsymbol{x})\right) = \int q(\boldsymbol{x}) \log \frac{q(\boldsymbol{x})}{p(\boldsymbol{x})} d\boldsymbol{z}. \tag{29}$$

Based on the above notation, the log-likelihood $\log \pi_{\boldsymbol{\theta}}(\boldsymbol{a}_i|\boldsymbol{s}_i)$ can be written as follows:

$$\log \pi_{\boldsymbol{\theta}}(\boldsymbol{a}_i|\boldsymbol{s}_i) = \int q_{\boldsymbol{\phi}}(\boldsymbol{z}|\boldsymbol{s}_i, \boldsymbol{a}_i) \log \pi_{\boldsymbol{\theta}}(\boldsymbol{a}_i|\boldsymbol{s}_i) d\boldsymbol{z} \tag{30}$$

$$= \int q_{\boldsymbol{\phi}}(\boldsymbol{z}|\boldsymbol{s}_i, \boldsymbol{a}_i) \left( \log \pi(\boldsymbol{a}_i|\boldsymbol{s}_i, \boldsymbol{z}) + \log p(\boldsymbol{z}|\boldsymbol{s}_i) - \log p(\boldsymbol{z}|\boldsymbol{s}_i, \boldsymbol{a}_i) \right) d\boldsymbol{z} \tag{31}$$

$$\begin{aligned}= & \int q_{\boldsymbol{\phi}}(\boldsymbol{z}|\boldsymbol{s}_i, \boldsymbol{a}_i) \log \frac{q(\boldsymbol{z}|\boldsymbol{s}_i, \boldsymbol{a}_i)}{p(\boldsymbol{z}|\boldsymbol{s}_i, \boldsymbol{a}_i)} d\boldsymbol{z} \\ & - \int q(\boldsymbol{z}|\boldsymbol{s}_i, \boldsymbol{a}_i) \log \frac{q(\boldsymbol{z}|\boldsymbol{s}_i, \boldsymbol{a}_i)}{p(\boldsymbol{z}|\boldsymbol{s}_i)} d\boldsymbol{z} \\ & + \int q(\boldsymbol{z}|\boldsymbol{s}_i, \boldsymbol{a}_i) \log \pi_{\boldsymbol{\theta}}(\boldsymbol{a}_i|\boldsymbol{s}_i, \boldsymbol{z}) d\boldsymbol{z} \end{aligned} \tag{32}$$

$$\begin{aligned}= & D_{\mathrm{KL}}\left(q(\boldsymbol{z}|\boldsymbol{s}_i, \boldsymbol{a}_i)||p(\boldsymbol{z}|\boldsymbol{s}_i, \boldsymbol{a}_i)\right) - D_{\mathrm{KL}}(q(\boldsymbol{z}|\boldsymbol{s}_i, \boldsymbol{a}_i)||p(\boldsymbol{z}|\boldsymbol{s}_i)) \\ & + \mathbb{E}_{\boldsymbol{z} \sim q(\boldsymbol{z}|\boldsymbol{s}_i, \boldsymbol{a}_i))} \left[\log \pi_{\boldsymbol{\theta}}(\boldsymbol{a}_i|\boldsymbol{s}_i, \boldsymbol{z})\right]. \end{aligned} \tag{33}$$

 In the first line, we consider marginalization over $\boldsymbol{z}$. As $\log \pi(\boldsymbol{a}|\boldsymbol{s})$ is independent of the latent variable $\boldsymbol{z}$, the equality in the first line holds. Because $D_{\mathrm{KL}}\left(q(\boldsymbol{z}|\boldsymbol{s}, \boldsymbol{a})||p(\boldsymbol{z}|\boldsymbol{s}, \boldsymbol{a})\right) > 0$, we can obtain a variant of the variational lower bound of the conditional log-likelihood:

$$\log \pi_{\boldsymbol{\theta}}(\boldsymbol{a}_i|\boldsymbol{s}_i) \geq -D_{\mathrm{KL}}(q_{\boldsymbol{\phi}}(\boldsymbol{z}|\boldsymbol{s}_i, \boldsymbol{a}_i)||p(\boldsymbol{z}|\boldsymbol{s}_i)) + \mathbb{E}_{\boldsymbol{z} \sim q(\boldsymbol{z}|\boldsymbol{s}_i, \boldsymbol{a}_i))} \left[\log \pi_{\boldsymbol{\theta}}(\boldsymbol{a}_i|\boldsymbol{s}_i, \boldsymbol{z})\right]. \tag{34}$$

Table 8: Effect of the dimensionality of the discrete latent variable. WK=walker2d.

| | infoDMPO $|Z| = 4$ | infoDMPO $|Z| = 8$ | infoDMPO $|Z| = 16$ | infoDMPO $|Z| = 32$ |
|---|---|---|---|---|
| pen-human-v0 | $75.7 \pm 18.9$ | $\mathbf{94.8 \pm 16.5}$ | $75.0 \pm 17.5$ | $86.7 \pm 12.4$ |
| WK-expert-v2 | $99.7\pm17.9$ | $\mathbf{112.1\pm0.4}$ | $108.8\pm6.8$ | $\mathbf{106.4\pm10.2}$ |
| WK-med.-expert-v2 | $89.1\pm25.7$ | $\mathbf{110.1\pm0.7}$ | $96.0\pm17.0$ | $\mathbf{109.9\pm0.6}$ |
| WK-med.-replay-v2 | $81.6\pm4.5$ | $\mathbf{86.7\pm3.2}$ | $85.4\pm3.7$ | $\mathbf{86.3\pm3.1}$ |
| WK-med.-v2 | $81.8\pm2.5$ | $\mathbf{85.0\pm0.8}$ | $69.9\pm28.3$ | $\mathbf{84.3\pm1.0}$ |

## C  Proof of contraction of the latent-max-Q operator

We consider operator $\mathcal{T}_{\boldsymbol{z}}$, which is given by

$$\mathcal{T}_{\boldsymbol{z}} Q(\boldsymbol{s}, \boldsymbol{a}) = \mathbb{E}_{\boldsymbol{s}'} \left[ r(\boldsymbol{s}, \boldsymbol{a}) + \gamma \max_{\boldsymbol{z}} Q(\boldsymbol{s}', \boldsymbol{\mu}(\boldsymbol{s}', \boldsymbol{z}')) \right]. \tag{35}$$

To prove the contraction of $\mathcal{T}_{\boldsymbol{z}}$, we use the infinity norm given by

$$\|Q_1 - Q_2\|_\infty = \max_{\boldsymbol{s} \in \mathcal{S}, \boldsymbol{a} \in \mathcal{A}} |Q_1(\boldsymbol{s}, \boldsymbol{a}) - Q_2(\boldsymbol{s}, \boldsymbol{a})|, \tag{36}$$

where $Q_1$ and $Q_2$ are different estimates of the Q-function. We consider the infinity norm of the difference between the two estimates, $Q_1$ and $Q_2$, after applying operator $\mathcal{T}_{\boldsymbol{z}}$:

$$\|\mathcal{T}_{\boldsymbol{z}} Q_1 - \mathcal{T}_{\boldsymbol{z}} Q_2\|_\infty \tag{37}$$

$$= \left| \mathbb{E}_{\boldsymbol{s}'} \left[ r(\boldsymbol{s}, \boldsymbol{a}) + \gamma \max_{\boldsymbol{z}'} Q_1(\boldsymbol{s}', \boldsymbol{\mu}(\boldsymbol{s}', \boldsymbol{z}')) \right] - \mathbb{E}_{\boldsymbol{s}'} \left[ r(\boldsymbol{s}, \boldsymbol{a}) + \gamma \max_{\boldsymbol{z}'} Q_2(\boldsymbol{s}', \boldsymbol{\mu}(\boldsymbol{s}', \boldsymbol{z}')) \right] \right| \tag{38}$$

$$= \left| \gamma \mathbb{E}_{\boldsymbol{s}'} \left[ \max_{\boldsymbol{z}'} Q_1(\boldsymbol{s}', \boldsymbol{\mu}(\boldsymbol{s}', \boldsymbol{z}')) \right] - \gamma \mathbb{E}_{\boldsymbol{s}'} \left[ \max_{\boldsymbol{z}'} Q_2(\boldsymbol{s}', \boldsymbol{\mu}(\boldsymbol{s}', \boldsymbol{z}')) \right] \right| \tag{39}$$

$$= \gamma \left| \mathbb{E}_{\boldsymbol{s}'} \left[ \max_{\boldsymbol{z}'} Q_1(\boldsymbol{s}', \boldsymbol{\mu}(\boldsymbol{s}', \boldsymbol{z}')) \right] - \mathbb{E}_{\boldsymbol{s}'} \left[ \max_{\boldsymbol{z}'} Q_2(\boldsymbol{s}', \boldsymbol{\mu}(\boldsymbol{s}', \boldsymbol{z}')) \right] \right| \tag{40}$$

$$= \gamma \left| \mathbb{E}_{\boldsymbol{s}'} \left[ \max_{\boldsymbol{z}'} Q_1(\boldsymbol{s}', \boldsymbol{\mu}(\boldsymbol{s}', \boldsymbol{z}')) - \max_{\boldsymbol{z}'} Q_2(\boldsymbol{s}', \boldsymbol{\mu}(\boldsymbol{s}', \boldsymbol{z}')) \right] \right| \tag{41}$$

$$\leq \gamma \left| \mathbb{E}_{\boldsymbol{s}'} \|Q_1 - Q_2\|_\infty \right| \tag{42}$$

$$\leq \gamma \|Q_1 - Q_2\|_\infty. \tag{43}$$

The above relationship shows the contraction of operator $\mathcal{T}_{\boldsymbol{z}}$.

## D  Effect of dimensionality of the discrete latent variable

In our evaluation, we first examined the effect of the dimensionality of the discrete latent variable. The results are presented in Table 8. As shown, infoDMPO with $|Z| = 8$ demonstrated the best performance, while the performance with $|Z| = 16$ and $|Z| = 32$ is comparable. These results show that the policy performance is not very sensitive to the dimensionality of the latent variable. However, the performance with $|Z| = 4$ is relatively weak, thereby indicating that the policy may not be sufficiently expressive when the dimensionality of the latent variable is significantly small. Because $|Z| = 8$ consistently provided satisfactory performance, $|Z| = 8$ was used in the subsequent evaluations.

## E  Comparison with additional baselines

We provide a comparison with additional baselines for the mujoco-v2 tasks in D4RL in Table 9. We present the results of MAPLE, which is a recent model-based offline algorithm that uses latent representations (Chen et al., 2021b). In addition, we provide the results of decision transformer (Chen et al., 2021a), as a representative transformer-based method. Although these methods are well-known and state-of-the-art, we focused on model-free and non-transformer-based methods in the main manuscript. For each baseline method, we adapted the results reported in the original paper. DMPO and infoDMPO provide a consistently better or comparable performance to these baseline methods, although our implementation of DMPO and infoDMPO does not employ techniques such as ensemble of critics. This result indicates a significant effect of the policy structure in offline RL.

Table 9: Results on mujoco tasks using D4RL-v2 datasets. Average normalized scores over the last 10 test episodes and five seeds are shown. HC = HalfCheetah, HP = Hopper, WK = Walker2d. The gray text indicates the performance lower than that of DMPO/infoDMPO. The bold text indicates the best performance.

| | | | MAPLE | Decision Transformer | DMPO | infoDMPO |
|---|---|---|---|---|---|---|
| | | | (paper) | (paper) | (ours) | (ours) |
| Med.-Expert | | HC | 63.5 ± 6.5 | 86.8 ± 1.3 | **91.1± 3.4** | **91.4± 2.5** |
| | | HP | 42.5 ± 4.1 | **107.6 ± 1.8** | 78.4± 19.0 | 94.5±14.9 |
| | | WK | 73.8 ± 8.0 | 108.1 ± 0.2 | **109.9± 0.4** | **110.1± 0.7** |
| Med.-Replay | | HC | **59.0 ± 0.6** | 36.6 ± 0.8 | 45.2± 0.8 | 46.7± 0.6 |
| | | HP | 87.5 ± 10.8 | 82.7 ± 7.0 | 89.2± 8.1 | **98.5± 2.0** |
| | | WK | 76.7 ± 3.8 | 66.6 ± 3.0 | 82.1± 3.8 | **86.7± 3.2** |
| Med. | | HC | **50.4 ± 1.9** | 42.6 ± 0.1 | 47.5± 0.4 | 48.6± 0.4 |
| | | HP | 21.1 ± 1.2 | 67.6 ± 1.0 | 71.2± 6.5 | **86.4± 7.6** |
| | | WK | 56.3 ± 10.6 | 74.0 ± 1.4 | 79.4± 4.7 | **85.0± 0.8** |

## F Hyperparameters and implementation details

**Computational resource and license** The experiments were run with Amazon Web Service and workstations with NVIDIA RTX 3090 GPUs and Intel Core i9-10980XE CPUs at 3.0 GHz. We used the physics simulator, MuJoCo (Todorov et al., 2012) under an institutional license, and later we switched to the Apache license.

**Software** The software versions used in the experiments are listed below:

- Python 3.8

- Pytorch 1.10.0

- Gym 0.21.0

- MuJoCo 2.1.0

- mujoco-py 2.1.2.14

We used the author-provided implementations for TD3+BC[2] and CQL[3]. DMPO, AWAC, mixAWAC, and IQL were implemented based on the the author-provided implementation of TD3. For IQL, we used the hyperparameters provided in Kostrikov et al. (2022). To minimize the difference between DMPO and AWAC, we used a delayed update of the policy in both DMPO and AWAC. For simplicity, we did not use a regularization technique for the actor such as the dropout layer used in (Kostrikov et al., 2022), although the use of such techniques should further improve the performance. In our implementation of DMPO, the value of z is a part of the input to the actor network. Thus, different behaviors corresponding to different values of $z$ are represented by the same actor network. The network architecture is illustrated in Figure 6.

**Computation of the advantage function** In DMPO, a policy is deterministic because both the gating policy $\pi(\boldsymbol{z}|\boldsymbol{s})$ and sub-policy $\pi(\boldsymbol{a}|\boldsymbol{s},\boldsymbol{z})$ are deterministic. Thus, the state-value function is given by

$$V^{\pi}(\boldsymbol{s}) = \max_{\boldsymbol{z}} Q^{\pi}(\boldsymbol{s}, \boldsymbol{\mu}(\boldsymbol{s},\boldsymbol{z})). \tag{44}$$

Therefore, the advantage function is given by

$$A^{\pi}(\boldsymbol{s},\boldsymbol{a}) = Q^{\pi}(\boldsymbol{s},\boldsymbol{a}) - V^{\pi}(\boldsymbol{s}) = Q^{\pi}(\boldsymbol{s},\boldsymbol{a}) - \max_{\boldsymbol{z}} Q^{\pi}(\boldsymbol{s},\boldsymbol{\mu}(\boldsymbol{s},\boldsymbol{z})). \tag{45}$$

In the policy update, we use the target actor in the second term in Equation 45. Thus, in our implementation, the advantage function is approximated as

$$A(\boldsymbol{s},\boldsymbol{a};\boldsymbol{w},\boldsymbol{\theta}') = Q(\boldsymbol{s},\boldsymbol{a};\boldsymbol{w}) - \max_{\boldsymbol{z}} Q(\boldsymbol{s},\boldsymbol{\mu}_{\boldsymbol{\theta}'}(\boldsymbol{s},\boldsymbol{z});\boldsymbol{w}). \tag{46}$$

---

[2]https://github.com/sfujim/TD3_BC
[3]https://github.com/young-geng/CQL

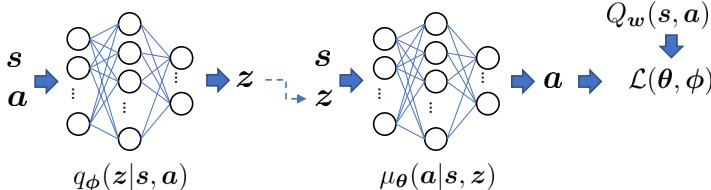

(a) Computation for maximizing $\mathcal{L}_{\mathrm{ML}}$ in Equation 11.

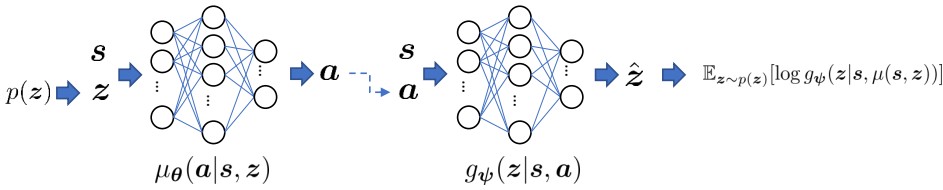

(b) Computation for maximizing $\sum_{i=1}^{N} \mathbb{E}_{\boldsymbol{z} \sim p(\boldsymbol{z})} \log g_{\boldsymbol{\psi}}(\boldsymbol{z}|\boldsymbol{s}_i, \boldsymbol{\mu}_{\boldsymbol{\theta}}(\boldsymbol{s}_i, \boldsymbol{z}))$.

Figure 6: Connection between $q_{\boldsymbol{\phi}}(\boldsymbol{z}|\boldsymbol{s}, \boldsymbol{a})$, $\mu_{\boldsymbol{\theta}}(\boldsymbol{s}, \boldsymbol{z})$, and $g_{\boldsymbol{\psi}}(\boldsymbol{z}|\boldsymbol{s}, \boldsymbol{a})$ during training.

**Target smoothing in DMPO** In DMPO, a policy is given by a mixture of deterministic sub-policies, where a sub-policy is selected in a deterministic manner, similar to that in Equation 3. Thus, the mixture policy in this framework is deterministic. As reported in Fujimoto & Gu (2021), the use of a deterministic policy may lead to overfitting of the critic to narrow peaks. Because our policy is deterministic, we also employed a technique called target policy smoothing used in TD3. Thus, the target value in Equation 17 is modified as follows:

$$y_i = r_i + \gamma \max_{\boldsymbol{z}' \in \mathcal{Z}} \min_{j=1,2} Q_{\boldsymbol{w}'_j}(\boldsymbol{s}', \boldsymbol{\mu}_{\boldsymbol{\theta}'}(\boldsymbol{s}', \boldsymbol{z}') + \epsilon_{\mathrm{clip}}), \tag{47}$$

where $\epsilon_{\mathrm{clip}}$ is given by

$$\epsilon_{\mathrm{clip}} = \min(\max(\epsilon, -c), c) \quad \text{where} \quad \epsilon \sim \mathcal{N}(0, \sigma), \tag{48}$$

and constant $c$ defines the range of the noise.

**Techniques for Antmaze tasks** In LAPO Chen et al. (2022), several techniques to stabilize the training of the value functions are used. Suppose the state-value function is approximated with $V_{\boldsymbol{w}_v}(\boldsymbol{s})$ parameterized with a vector $\boldsymbol{w}_v$, and the Q-function is approximated with two models, which are represented by $Q_{\boldsymbol{w}_j}(\boldsymbol{s}, \boldsymbol{a})$ for $j = 1, 2$. The state-value function is updated by minimizing

$$\mathcal{L}_v(\boldsymbol{w}_v) = \sum_{(\boldsymbol{s}_i, \boldsymbol{a}_i) \in \mathcal{D}} \|\tilde{y}_i - V_{\boldsymbol{w}_v}(\boldsymbol{s}_i)\|^2, \tag{49}$$

where the target value $\tilde{y}_i$ is the clipped target value computed as

$$\tilde{y}_i = \max\left(\min\left(y_i, v_{\max}\right), v_{\min}\right) \tag{50}$$

and $y_i$ is computed as

$$y_i = c \min_{j=1,2} Q_{\boldsymbol{w}_j}(\boldsymbol{s}_i, \boldsymbol{a}_i) + (1-c) \max_{j=1,2} Q_{\boldsymbol{w}_j}(\boldsymbol{s}_i, \boldsymbol{a}_i) \tag{51}$$

and $c$ is a constant, and we used $c = 0.7$ as in Chen et al. (2022). The minimum and maximum target value $v_{\min}$ and $v_{\max}$ are computed as

$$v_{\min} = \frac{1}{1-\gamma} \min_{r_i \in \mathcal{D}} r_i \tag{52}$$

$$v_{\max} = \frac{1}{1-\gamma} \max_{r_i \in \mathcal{D}} r_i. \tag{53}$$

The Q-function is updated by minimizing the following objective function:

$$\mathcal{L}_q(\boldsymbol{w}) = \sum_{(\boldsymbol{s}_i, \boldsymbol{a}_i, r_i, \boldsymbol{s}_i') \in \mathcal{D}} \left\| r_i + \gamma V_{\boldsymbol{w}_v}(\boldsymbol{s}_i') - Q_{\boldsymbol{w}}(\boldsymbol{s}_i, \boldsymbol{a}_i) \right\|^2. \tag{54}$$

For the antmaze tasks, we also used same techniques in DMPO.

**Implementation of mixAWAC**  The difference between mixAWAC and AWAC is a policy representation. For mixAWAC, we used a Gaussian mixture policy. The discrete latent variable is sampled from a categorical distribution, and the corresponding Gaussian component policy is used to sample the action. As in DMPO, the latent variable is represented as a one-hot vector, and the neural network that represents the Gaussian components takes the state and the one-hot vector as its input. The key part of the implementation is how to sample from a categorical distribution in a differentiable manner. We used the Gumbel-max trick for this purpose (Chris J. Maddison, 2014; Jang et al., 2017; Maddison et al., 2017). The Gumbel-max trick is often used to learn discrete latent variable in VAE (Kingma & Welling, 2014).

In our implementation, the activation function of the last layer of the gating policy is the softmax function. The discrete latent variable is sampled using on the Gumbel-max trick based on the output of the gating policy.

**Implementation of LP-AWAC**  As in our implementation of AWAC, mixAWAC, and DMPO, the double-clipped Q-learning is employed in LP-AWAC. In additioned to the Q-function, the state value function $V_{\boldsymbol{w}}(\boldsymbol{s})$ is trained by minimizing the mean squared error:

$$\mathcal{L}_{\text{LP-AWAC}}(\boldsymbol{w}) = \sum_{(\boldsymbol{s}_i, \boldsymbol{a}_i) \in \mathcal{D}} \left\| V_{\boldsymbol{w}}(\boldsymbol{s}_i) - \min_{j=1,2} Q_{\boldsymbol{w}_j}(\boldsymbol{s}_i, \boldsymbol{a}_i) \right\|_2^2 \tag{55}$$

In LP-AWAC, the conditional VAE is trained using advantage weighting. Denoting the approximated posterior and likelihood by $q_{\boldsymbol{\phi}}(\boldsymbol{z}|\boldsymbol{s}, \boldsymbol{a})$ and $p_{\boldsymbol{\psi}}(\boldsymbol{a}|\boldsymbol{s}, \boldsymbol{z})$, respectively, the encoder and decoder are trained by maximizing the following objective function:

$$\mathcal{L}_{\text{cvae}}(\boldsymbol{\phi}, \boldsymbol{\psi}) = \sum_{(\boldsymbol{s}_i, \boldsymbol{a}_i, r_i, \boldsymbol{s}_i') \in \mathcal{D}} W(\boldsymbol{s}_i, \boldsymbol{a}_i) \left( -D_{\text{KL}}(q_{\boldsymbol{\phi}}(\boldsymbol{z}|\boldsymbol{s}_i, \boldsymbol{a}_i)||p(\boldsymbol{z}|\boldsymbol{s}_i)) + \mathbb{E}_{\boldsymbol{z} \sim q(\boldsymbol{z}|\boldsymbol{s}_i, \boldsymbol{a}_i)} \left[ \log p_{\boldsymbol{\psi}}(\boldsymbol{a}_i|\boldsymbol{s}_i, \boldsymbol{z}) \right] \right), \tag{56}$$

where $W(\boldsymbol{s}_i, \boldsymbol{a}_i)$ is the weight for advantage weighting. In our experiments, we used the normalized advantage weighting in Equation 19. Then, the deterministic latent actor $\boldsymbol{\mu}_{\boldsymbol{\theta}}(\boldsymbol{s})$ is trained to output the latent variable $\boldsymbol{z}$ by maximizing the expected Q-value:

$$\mathcal{L}_{\text{latent-actor}}(\boldsymbol{\theta}) = \sum_{(\boldsymbol{s}_i, \boldsymbol{a}_i) \in \mathcal{D}} Q_{\boldsymbol{w}_1}(\boldsymbol{s}_i, g_{\boldsymbol{\psi}}(\boldsymbol{s}_i, \boldsymbol{\mu}_{\boldsymbol{\theta}}(\boldsymbol{s}_i))), \tag{57}$$

where $g_{\boldsymbol{\psi}}(\boldsymbol{s}, \boldsymbol{a})$ is the decoder. The objective function for learning the continuous latent variable in LP-AWAC is very similar to that of DMPO in Equation 11 for learning the discrete latent variable. When considering the deterministic latent actor $\boldsymbol{\mu}_{\boldsymbol{\theta}}(\boldsymbol{s})$ in LP-AWAC as the gating policy that approximately solves $\arg\max_{\boldsymbol{z}} Q(\boldsymbol{s}, \boldsymbol{z})$, LP-AWAC can be considered as the variant of DMPO using the continuous latent variable. Thus, the difference between DMPO and LP-AWAC indicates the difference of the discrete and continuous latent variable in our framework.

Table 10: Hyperparameters of DMPO & infoDMPO.

| | Hyperparameter | Value |
|---|---|---|
| **Hyperparameters** | Optimizer | Adam |
| | Critic learning rate | 3e-4 (mujoco-v2, adroit) / 2e-4 (Antmaze) |
| | Actor learning rate | 3e-4 (mujoco-v2, adroit) / 2e-4 (Antmaze) |
| | Posterior learning rate | 3e-4 (mujoco-v2, adroit) / 2e-4 (Antmaze) |
| | Mini-batch size | 256 |
| | Discount factor | 0.99 |
| | Target update rate | 5e-3 |
| | Policy noise | 0.2 |
| | Policy noise clipping | (-0.5, 0.5) |
| | Policy update frequency | 2 |
| **Architecture** | Critic hidden dim | 256 |
| | Critic hidden layers | 2 (mujoco-v2, adroit) / 3 (Antmaze) |
| | Critic activation function | ReLU |
| | Actor hidden dim | 256 |
| | Actor hidden layers | 2 (mujoco-v2, adroit) / 3 (Antmaze) |
| | Actor activation function | ReLU |
| | Posterior hidden dim | 256 |
| | Posterior hidden layers | 2 (mujoco-v2, adroit) / 3 (Antmaze) |
| | Posterior activation function | ReLU |
| **DMPO** | Score scaling $\alpha$ | 5.0 (human, Antmaze) |
| | | 10.0 (mujoco-v2) |
| **infoDMPO** | learning rate of the posterior for infomax | 3e-6 (Adroit) |
| | | 5e-7 (mujoco-v2) |
| | | 5e-7 (Antmaze) |
| | Score scaling $\alpha$ | 5.0 (Antmaze, HP-med.-expert) |
| | | 10.0 (others) |

**Number of updates** In the pen-human-v0, hammer-human-v0, door-human-v0, and relocate-human-v0 tasks, the number of samples contained in the dataset is significantly smaller than that for the other datasets. While the datasets for mujoco tasks contained approximately 1 million samples, the numbers of samples in the adroit-human tasks were as follows: pen-human-v0: 4,950 samples, hammer-human-v0: 11,264 samples, door-human-v0: 6,703 samples, and relocate-human-v0: 9,906 samples. Thus, in the pen-human-v0, hammer-human-v0, door-human-v0, and relocate-human-v0 tasks, we updated the policy 10,000 times, whereas for the other tasks, we updated the policy 1 million times. The aforementioned number of policy updates was applied to all methods.

**Hyperparameters** Tables 10–14 provide the hyperparameters used in the experiments. Regarding $\lambda$ in infoDMPO, the first and second terms in Equation (13) are maximized separately. Thus, we implicitly set the value of $\lambda$ by setting the different learning rates for the first and second terms in Equation (13). The learning rate for the first term in Equation (13) was fixed to 3e-4. We tested a set of the learning rate {1e-7, 5e-7, 1e-6, 3e-6} for the second term in Equation (13), and we reported the best results in the paper.

Table 11: Hyperparameters of AWAC.

|  | Hyperparameter | Value |
|---|---|---|
| **Hyperparameters** | Optimizer | Adam |
|  | Critic learning rate | 3e-4 |
|  | Actor learning rate | 3e-4 |
|  | Mini-batch size | 1024 |
|  | Discount factor | 0.99 |
|  | Target update rate | 5e-3 |
|  | Policy update frequency | 2 |
|  | Score scaling $\alpha$ | 10.0 |
| **Architecture** | Critic hidden dim | 256 |
|  | Critic hidden layers | 2 |
|  | Critic activation function | ReLU |
|  | Actor hidden dim | 256 |
|  | Actor hidden layers | 2 |
|  | Actor activation function | ReLU |

Table 12: Hyperparameters of TD3+BC. The default hyperparameters in the TD3+BC GitHub are used.

|  | Hyperparameter | Value |
|---|---|---|
| **Hyperparameters** | Optimizer | Adam |
|  | Critic learning rate | 3e-4 |
|  | Actor learning rate | 3e-4 |
|  | Mini-batch size | 256 |
|  | Discount factor | 0.99 |
|  | Target update rate | 5e-3 |
|  | Policy noise | 0.2 |
|  | Policy noise clipping | (-0.5, 0.5) |
|  | Policy update frequency | 2 |
|  | $\alpha$ | 2.5 |
| **Architecture** | Critic hidden dim | 256 |
|  | Critic hidden layers | 2 |
|  | Critic activation function | ReLU |
|  | Actor hidden dim | 256 |
|  | Actor hidden layers | 2 |
|  | Actor activation function | ReLU |

Table 13: Hyperparameters of CQL. The default hyperparameters in the CQL GitHub are used.

| | Hyperparameter | Value |
|---|---|---|
| Hyperparameters | Optimizer | Adam |
| | Critic learning rate | 3e-4 |
| | Actor learning rate | 3e-5 |
| | Mini-batch size | 256 |
| | Discount factor | 0.99 |
| | Target update rate | 5e-3 |
| | Target entropy | -1·Action Dim |
| | Entropy in Q target | False |
| | Lagrange | False |
| | $\alpha$ | 10 |
| | Pre-training steps | 40e3 |
| | Num sampled actions (during eval) | 10 |
| | Num sampled actions (logsumexp) | 10 |
| Architecture | Critic hidden dim | 256 |
| | Critic hidden layers | 3 |
| | Critic activation function | ReLU |
| | Actor hidden dim | 256 |
| | Actor hidden layers | 3 |
| | Actor activation function | ReLU |

Table 14: Hyperparameters of IQL. The default hyperparameters in the IQL in the paper Kostrikov et al. (2022) are used.

| | Hyperparameter | Value |
|---|---|---|
| IQL hyperparameters | Optimizer | Adam |
| | Critic learning rate | 3e-4 |
| | Actor learning rate | 3e-4 |
| | Mini-batch size | 256 |
| | Discount factor | 0.99 |
| | Target update rate | 5e-3 |
| | Expectile | 0.7 (mujoco-v2) |
| | | 0.7 (adroit) |
| | | 0.9 (antmaze) |
| | Advantage scale | 3.0 (mujoco-v2) |
| | | 0.5 (adroit) |
| | | 10.0 (antmaze) |
| | Actor learning rate scheduling | cosine |
| Architecture | Critic hidden dim | 256 |
| | Critic hidden layers | 2 |
| | Critic activation function | ReLU |
| | Actor hidden dim | 256 |
| | Actor hidden layers | 3 |
| | Actor activation function | ReLU |

