# OpenReview forum: "Offline Reinforcement Learning with Mixture of Deterministic Policies"
_TMLR — Accepted by TMLR_

### Review · Reviewer_PB5R · 2023-07-10

**Summary Of Contributions:**

The paper deals with offline RL.
It is hypothesized that the use of a multitude of deterministic policies can help to reduce the extrapolation error of the value function.
An algorithm in two variants is presented that learns a gating agent and a multitude of deterministic policies.
Both variants are tested on a number of deterministic benchmarks and show very good, partly superior results compared to established methods.

**Audience:**

Yes

**Claims And Evidence:**

Yes

**Requested Changes:**

From my point of view, no major changes are necessary.

A few small remarks:

* Regarding "a one-step offline RL approach" it should be considered whether [1] should not be cited as well, in which as far as I know this idea was presented for the first time.\
[1] C. Gulcehre et al., Addressing Extrapolation Error in Deep Offline Reinforcement Learning, 2021
* equation 1 -> Equation 1
* 3.0GHZ -> 3.0 GHz


In the References
* variationalapproach ->  variational approach
* Offline model-base adaptable policy learning. -> Offline model-based adaptable policy learning.
* In some places, there are unintentional, misspelled lowercase letters: Why should i trust you, bellman? the bellman error is a poor replacement for value error, gumbel, fisher


**Strengths And Weaknesses:**

**Strengths**
* The paper is well written.
* It investigates an interesting direction for improving offline RL.
* The results are promising.

**Weaknesses**
* Whether the assumption that serves as motivation for the approach is really confirmed remains unclear.

---

> ### Author Response · Authors · 2023-07-11
> **Response to the comments from Reviewer PB5R**
>
> We appreciate the reviewer's efforts to evaluate our manuscripts. According to the comments from the reviewer, we revised the following points.
>
> - Changed "equation x" to "Equation x"
> - in page 17, "3.0GHz" -> "3.0 GHz"
> - Modified the misspelling and unintentional lowercase letters in the references
>
> Regarding the first comment on one-step RL, we did not clearly understand what was meant by  " it should be considered whether [1] should not be cited as well.." We think the reference for [1] is probably missing in the comment.
> We would appreciate it if the reviewer provides some clarification.

---

> > ### Comment · Reviewer_PB5R · 2023-07-11
> > **Missing reference**
> >
> > `We think the reference for [1] is probably missing in the comment` You are absolutely right, I had forgotten to give the reference. I have now added it to the review.
> >
> > [1] C. Gulcehre et al., Addressing Extrapolation Error in Deep Offline Reinforcement Learning, 2021

---

> > > ### Author Response · Authors · 2023-07-12
> > > **Thank you for clarification**
> > >
> > > Thank you for providing the reference for the one-step RL. We added Gulcehre et al. (2020) as a reference and updated the manuscript.
> > > If you have any further comments, please let us know.

---

### Review · Reviewer_LayE · 2023-07-24

**Summary Of Contributions:**

This paper proposes a new offline RL algorithm called, Deterministic Mixture Policy Optimization (DMPO), to overcome the issue of most existing methods that fit a unimodal distribution (typically Gaussian distribution) when the offline data distribution is potentially multimodal.

The key idea of DMPO is to leverage a mixture policy built on a gating policy and multiple sub-policies. DMPO learns discrete latent variables to separate the state-action space based on the offline dataset via a typical conditional VAE manner. So that the gating policy selects a discrete variable $z$ under a specific state and the multiple sub-policies are represented by a $z$-conditioned deterministic policy.

The policy optimization objective is derived from a variational lower bound of typical policy optimization objective. In addition, a Mutual-Information (MI) objective is introduced to encourage the non-overlapness of multiple sub-policies. The theorems on contraction and convergence are provided under classic conditions.

DMPO is evaluated in a toy example to show the effectiveness in dealing with synthesized multimodal data, while also compared against common baselines in D4RL, Antmaze and Adroit. Other analysis and visualization are provided.

**Audience:**

Yes

**Claims And Evidence:**

No

**Requested Changes:**

1. Can the authors provide a pros-cons comparison between DMPO and Diffusion-QL [1], from the methodology perspective? I suggest the authors to include Diffusion-QL in the empirical comparison as well.
2. Can the authors explain the gap existing in deriving the policy optimization objective with a stochastic policy while using a deterministic policy in DMPO?
3. Can the authors explain the rationale of choosing Eq.9 as the prior?
4. How $z$ are represented in practice since it should be taken as input in $u(s,z)$, one-hot representation or some learned representation by techniques like VQ-VAE?
5. For infoDMPO, how is the hyperparameter $\lambda$ selected?
6. I recommend the authors to include the concrete implementation of mixAWAC in the appendix.

**Strengths And Weaknesses:**

### Strengths:

- The motivation of this work is clear. I appreciate the expression in Figure 1.
- The organization of method presentation is good, including both the derivation, theory and practical implementation.
- The experiments are multifaceted.

&nbsp;

### Weaknesses:

- Although the motivation of this paper is clear and makes sense to me, it is not novel since a prior work [1] studies the same problem. Therefore, the work[1] is closely related to this paper, while it is not included in this paper for discussion or empirical comparison.
- My major concern lies at the introduction of the policy optimization objective in Section 4.1.
    - I feel a gap between the derivation based on a stochastic policy and the deterministic policy used in DMPO. I found this is implicitly done by making use of MSE implementation of the log-likelihood maximization.
    - In addition, I cannot understand the rationale of choosing Eq.9 as the prior distribution. I think more explanation may help.

&nbsp;

### Minors:
- In Eq.12, the sampling distributions for the expectation notations are inconsistent.

---
### Reference:

[1] Wang et al., Diffusion Policies as an Expressive Policy Class for Offline Reinforcement Learning. ICLR 2023

---

> ### Author Response · Authors · 2023-07-27
> **Response to the comments from Reviewer LayE**
>
> Thank you for your constructive comments. We summarize our response as follows. We will update the manuscript accordingly after the experiments with Diffusion QL are finished.
>
> > 1. Can the authors provide a pros-cons comparison between DMPO and Diffusion-QL [1], from the methodology perspective? I suggest the authors to include Diffusion-QL in the empirical comparison as well.
>
> The fundamental difference between Diffusion QL and DMPO comes from the choice of the policy structure: DMPO employs a categorical VAE-like policy structure [1], and Diffusion QL employs a diffusion-model-like policy structure. Theoretically, both VAE and diffusion models are based on the variational lower bound of the log-likelihood, and it is possible to develop a variant of DMPO using the diffusion model. The advantage of Diffusion QL over DMPO is that Diffusion QL can train a highly expressive policy. On the other hand, the advantage of DMPO over Diffusion QL is computational cost. The process of noising and denoising samples during the training of a diffusion model is computationally expensive. As detailed in the paper, the training time for Diffusion QL is comparable to CQL. In our preliminary experiment, we found that the training time for CQL is significantly longer than that for DMPO. Therefore, we expect that the wall-clock time to train a policy with Diffusion QL will be also significantly longer than that with DMPO. We will evaluate the wall-clock time for training with Diffusion QL and DMPO.
>
> Furthermore, inference in Diffusion QL is computationally expensive. In Diffusion QL, N denoising steps are required to generate one action. When N=5, 5 forward passes of the policy network are required. In addition, to sample a good action, Diffusion QL needs to sample multiple actions and select the best one. Specifically, the author-provided implementation of Diffusion QL samples 50 actions and uses the one with the maximum expected Q-value as the best action. In contrast, DMPO generates one action for each sub-policy. For instance, if the number of sub-policies is 8, then 8 actions are generated, and the one with the maximum expected Q-value is chosen as the best action. To generate one action for each sub-policy, one forward pass of the policy network is necessary. Therefore, the computational cost for inference is significantly lower in DMPO compared with Diffusion QL, and this point will be an advantage of DMPO when a policy needs to be used in systems where inference should be fast.
>
> Regarding the empirical comparison, we are currently running the author-provided codes in our machines. We will report the empirical comparison with Diffusion QL when the experiment is finished. For now, judging from the results reported in [1], the performance of DMPO is comparable to Diffusion QL on benchmark tasks such as locomotion tasks with Gym and Antmaze tasks. While DMPO may not outperform Diffusion QL, we believe that the contributions of this paper still hold because our empirical result demonstrates the effect of a policy structure on offline RL, and DMPO has some advantages over Diffusion QL such as the computational cost.
>
> [1] E. Jang et al., Categorical Reparameterization with Gumbel-Softmax, ICLR 2017.
>
> > 2. Can the authors explain the gap existing in deriving the policy optimization objective with a stochastic policy while using a deterministic policy in DMPO?
>
> When training a stochastic policy, the first term in Equation (7) can be directly maximized because it is trivial to compute the log-likelihood $\log \pi (a|s)$. However, when a policy is given by a mixture of deterministic policies, it is not trivial. For this reason, we used the variational lower bound in Equation (8). In addition, as the reviewer indicated, $\mathbb{E}[\log \pi(a|s,z)]$ is replaced with MSE, which is the approximation used in VAE. As described in [2], the use of the objective in Equation (8) instead of $\mathbb{E}[\log \pi(a|s)]$ leads to the approximation gap, $\mathbb{E}[\log \pi(a|s)] - \ell_{\textrm{cvae}}(s, a)$ as in VAE.
> Addressing the gap using techniques such as the one proposed in [2] may further improve the performance of DMPO.
>
> [2] C. Cremer et al., Inference Suboptimality in Variational Autoencoders, ICML 2018.
>
> > 3. Can the authors explain the rationale of choosing Eq.9 as the prior?
>
> In Equation 8, $p(z|s)$ is the distribution induced by executing the policy $\pi$. In our framework, $p(z|s)$ should be the gating policy $\pi_{\textrm{gate}}(z|s)$, which determines the latent variable as $z = \arg \max_{z’} Q_w (s, \mu(s, z’))$. However, the gating policy is not explicitly modeled in our framework because it would introduce computation complexity. To approximate the gating policy represented by the arg max function over the Q-function, we used the Softmax distribution, which is often used to approximate the arg max function in machine learning.

---

> > ### Author Response · Authors · 2023-07-27
> > **Response to the comments from Reviewer LayE**
> >
> > > 4. How $z$ are represented in practice since it should be taken as input in $\mu(s,z)$, one-hot representation or some learned representation by techniques like VQ-VAE?
> >
> > Z is represented as a one-hot vector in our implementation. For reparametrization, we used the Gumbel-Max trick [1,3,4]. It would be possible to extend our framework by learning the discrete representations as in VQ-VAE. Such extensions are left for future work.
> >
> > [3] C. J. Maddison et al., A* sampling, NeurIPS 2014.
> >
> > [4] C. J. Maddison et al.,  The Concrete Distribution: A Continuous Relaxation of Discrete Random Variables, ICLR 2016.
> >
> > > 5. For infoDMPO, how is the hyperparameter $\lambda$ selected?
> >
> > As indicated in Algorithm 1, the first and second terms in Equation (13) are maximized separately. Thus, we implicitly set the value of $\lambda$ by setting the different learning rates for the first and second terms in Equation (13). The learning rate for the first term in Equation (13) was fixed to 3e-4. We tested a set of the learning rate {1e-7, 5e-7, 1e-6, 3e-6} for the second term in Equation (13), and we reported the best results in the paper. The values of the learning rates are provided in Table 9 in Appendix F.
> >
> > > 6. I recommend the authors to include the concrete implementation of mixAWAC in the appendix.
> >
> > The difference between mixAWAC and AWAC is a policy representation. For mixAWAC, we used a Gaussian mixture policy. The discrete latent variable is sampled from a categorical distribution, and the corresponding Gaussian component policy is used to sample the action.
> > In mixAWAC, the latent variable is represented as a one-hot vector, and the neural network that represents the Gaussian components takes the state and the one-hot vector as its input. The key part of the implementation is how to sample from a categorical distribution in a differentiable manner. We used the Gumbel-Max trick for this purpose. We will include the implementation details of mixAWAC in the appendix when we update the manuscript.
> >
> > To clarify the above points, we will update the manuscript accordingly after the experiments with Diffusion QL are done.  If you have any further comments, please let us know.

---

> > > ### Author Response · Authors · 2023-08-02
> > > **Comparison with Diffusion QL**
> > >
> > > As a follow-up, we report the result of running the author-provided implementation of Diffusion QL in our computational environment.
> > >
> > > The results of Diffusion QL and its comparison with DMPO and infoDMPO are summarized as follows:
> > >
> > > ||| Diffusion QL | DMPO | infoDMPO |
> > > | --- | --- | ---- | ---- | ---- |
> > > |*expert*| HC | 86.3±15.9 | **97.0±1.0** | 95.6±2.0 |
> > > |*expert*| HP | 84.3±24.2 | 93.6±15.1 | **107.5±2.9** |
> > > |*expert*| WK |109.0±0.6 | 111.4±0.3 | **112.1±0.4** |
> > > |*med.-e.*| HC | 83.8±15.3 | **91.1±3.4** | **91.4±2.5** |
> > > |*med.-e.*| HP| 88.1±25.7 | 78.4±19.0 | **94.5±14.9** |
> > > |*med.-e.*| WK | **110.1±0.6** | **109.9±0.4** | **110.1±0.7** |
> > > |*med.-r.*| HC |  45.6±0.6 | 45.2±0.8 | **46.7±0.6** |
> > > |*med.-r.*| HP| 56.1±24.0 | 89.2±8.1| **98.5±2.0** |
> > > |*med.-r.*| WK | **84.1±17.0** | 82.1±3.8 | **86.7±3.2** |
> > > |*med.*| HC | 46.7±0.7 | 47.5±0.4 | **48.6±0.4** |
> > > |*med.*| HP|  57.1±11.4 | 71.2± 6.5 | **86.4±7.6** |
> > > |*med.*| WK |62.1±20.6 | 79.4±4.7 | **85.0±0.8** |
> > > |*rnd.*| HC | **17.5±0.2** | 15.8±1.6 | 16.3±1.2 |
> > > |*rnd.*| HP| 7.8±0.5 | 12.0±10.0 | **20.4±9.8** |
> > > |*rnd.*| WK | **6.2±3.4** | 2.5± 2.6 | 2.3±2.0 |
> > >
> > >
> > > | | Diffusion QL | DMPO | infoDMPO |
> > > | --- | --- | ---- | ---- |
> > > |umaze| 80.4±35.3 | **92.8±2.1** | 89.4±5.1 |
> > > |umaze-d|8.0±21.9 | 32.6±25.6 | **34.8±18.0** |
> > > |med.-p.| **60.5±48.8** | **63.0±13.0** | **62.6±6.8**|
> > > |med.-d.|12.4±30.0 | 75.0±8.5 | **82.8±4.4**|
> > > |large-p.| **44.4±48.5** | **42.2±23.0** | **47.4±14.5**|
> > > |large-d.| **48.6±48.8** | **56.6±4.5** | 38.0±4.8|
> > >
> > > The results reported above are based on the offline model selection recommended in the paper.
> > > While the performance of Diffusion QL is comparable to DMPO and infoDMPO on antmaze tasks, the overall performance of DMPO and infoDMPO on mujoco locomotion tasks was better than that of Diffusion QL in our experiment.
> > >
> > > We also evaluated the computational time of Diffusion QL and infoDMPO.
> > >
> > > Action inference time was as follows:
> > > - Diffusion QL: 4.0ms to 4.3ms
> > > - infoDMPO: 1.3ms
> > >
> > > The time for training with 1 million steps was as follows:
> > > - Diffusion QL: 600min
> > > - infoDMPO:170min
> > >
> > > We used a workstation with GPU RTX A6000 and CPU Core™ i9-10980XE for this evaluation. These results show the computational cost of infoDMPO is significantly lower than that of Diffusion QL.
> > >
> > > We will add these results to the manuscript.
> > > We hope that our response resolves the reviewer's concerns. If you have any further comments, please let us know.

---

### Review · Reviewer_DypM · 2023-07-27

**Summary Of Contributions:**

In the offline reinforcement learning (RL) setting, it has been observed that querying the value of out-of-distribution (OOD) actions may cause an accumulation of bootstrapping errors. To address this issue, existing works may penalize deviation from behavior policy or underestimate OOD actions.

This work addresses the aforementioned issue from the perspective of the parameterization of policies. Prior works often use (squashed) Gaussian policies, which admit a single-mode behavior, while the data are usually multi-modal. Using a Gaussian policy to model multi-modal data might lead to interpolation between modes, causing evaluations of OOD actions, hence the accumulation of bootstrapping error.

From this aspect, the authors propose to divide the state-action space by learning discrete latent variables that activate (deterministic) sub-policies, to avoid interpolation between modes. And using this hierarchical policy structure to address the evaluation of OOD actions is novel to the best of my knowledge.

The authors then provide empirical evaluation in D4RL locomotion and AntMaze environments along with a toy multi-modal environment, comparing DMPO to several offline RL algorithms.

**Audience:**

Yes

**Claims And Evidence:**

Yes

**Requested Changes:**

- If the authors agree with the weaknesses point 1 and 2, the authors may need carefully change their claims to make them more precise.

- It is preferable to see further analysis of why discrete latent variables are better choices than continuous ones, as I found the current empirical comparison versus LAPO- could be made more convincing.

Minor points:

- ``In addition, DMPO outperformed LAPO-, which is the state-of-the-art method in offline RL." It is arguable that LAPO- can be still considered SOTA when several components of LAPO have been removed.

**Strengths And Weaknesses:**

**Strengths**

- This paper is overall well-written and easy to follow.

- It is true that Gaussian policy would lead to extrapolation between modes, hence using a mixture of policies could be a solution.

- Using the hierarchical design to avoid evaluation of OOD actions is novel to the best of my knowledge.

- Empirical results appeared to be strong although some improvements could still be made.

**Weaknesses**

- The motivating hypothesis might not be necessarily correct. In the first paragraph of section 1, the authors stated: `` Our hypothesis is that the evaluation of the out-of-distribution actions can be avoided by dividing the state-action space." The evaluation of OOD actions is not necessarily solved by separating offline action modes. Taking a dummy MDP example, suppose $\mathcal{S} = \{s\}$ and $\mathcal{A} = \{a_1, a_2\}$ and both actions leads to a self-loop to $s$, and rewards corresponding to the actions are $r_1=-1, r_2=-2$ respectively. Suppose the $Q$-function is initialized to be $Q(s, a_1) = Q(s, a_2) = 0$. Considering an offline dataset contains only $a_1$, training an vanilla RL agent (without further regularization)  with this dataset will still lead to evaluations of $a_2$ even though there is only one mode in the dataset, because of the over-estimation of $a_2$.

- Although I found the motivation hypothesis might not be fully correct, using latent policy to divide the state-action space should be able to avoid evaluation of OOD actions, because the latent policy avoids generating OOD actions, as observed in LAPO [1]. However, the OOD action issue might be mainly addressed by using latent policy rather than dividing the state-action space. The claims could be made more precises as I believe dividing state-action space without using latent policy won't be sufficient.

- In particular, LAPO has already provided a promising solution to avoid evaluations of OOD actions via latent policy. According to Figure 2 in [1], using latent policy alone could learn multimodal behaviors and avoid extrapolation between modes.
Although the authors provided comparisons between using discrete latent variables and continuous ones in Section 7, it is somewhat difficult to understand whether the difference is caused by the choice of latent variable or practical implementation.
It would be more compelling if the authors could also give some toy examples to demonstrate the advantage of discrete latent variables.


- The comparison versus LAPO- (LAPO minus) is not quite convincing. I understand that techniques such as action normalization and target clipping could significantly change the training dynamics and final performance. However, removing components of LAPO may not result in a fair comparison as DMPO also introduces techniques such as advantage normalization in Eq (18).


**Questions**

- ``While maximizing the objective $ L_\mathrm{ML} $ in Equation 11, both the actor $ µ_\theta (s, z)$ and auxiliary distribution $g_\psi (z|s, a)$ are updated.'' $ L_\mathrm{ML} $ appears to be an function of $\theta$ and $\phi$, I wonder how is $g_\psi$ updated when maximizing $L_\mathrm{ML} $.

[1] Chen, Xi, et al. "Latent-variable advantage-weighted policy optimization for offline rl." arXiv preprint arXiv:2203.08949 (2022).

---

> ### Author Response · Authors · 2023-08-02
> **Response to Reviewer DypM**
>
> Thank you for evaluating our manuscript. We think your valuable comments will significantly improve the quality of our manuscript. Our response is summarized as follows.
> As we are running experiments to address the concern raised by the reviewer, Section 7 will be updated after the experiments are completed.
>
> > "While maximizing the objective  in Equation 11, both the actor $\mu(s,z)$ and auxiliary distribution $g_{\psi} (z|s,a)$ are updated." $\mathcal{L}_{\textrm{ML}}$ appears to be an function of $\theta$ and $\phi$, I wonder how is $g$ updated when maximizing.
>
> Thank you for pointing this out. We found that the posterior $q(z|s,a)$ and the auxiliary distribution $g_{\psi} (z|s,a)$ are switched in the description.
>
> The correct description should be like this:
>
> "While maximizing the objective $\mathcal{L}_{\textrm{ML}}$ in Equation 11, both the actor $\mu(s,z)$ and posterior distribution $q(z|s,a)$ are updated; However, the auxiliary distribution $g(z|s,a)$ is frozen."
>
> Due to the limitation of the markdown in OpenReview, we omit $\theta$ and $\psi$ in the above sentence. Please take a look at the updated manuscript for a precise description.
>
> > Considering an offline dataset contains only $a_1$, training an vanilla RL agent (without further regularization) with this dataset will still lead to evaluations of $a_2$ even though there is only one mode in the dataset, because of the over-estimation of $a_2$.
>
> > If the authors agree with the weaknesses point 1 and 2, the authors may need carefully change their claims to make them more precise.
>
> Thank you for the comments. We agree with the reviewer. The problem indicated by the reviewer cannot be addressed by our approach.
> We do not intend to claim that dividing the state-action space can perfectly avoid generating OOD action in offline RL.
> We claim that dividing the state-action space can mitigate the problem of generating OOD action when the data distribution in a given dataset is multimodal.
> As the reviewer indicated, if the dataset only covers a subset of actions, there is the potential to overestimate the value of actions not contained in the dataset. In such cases, dividing the state action space does not solve the problem of generating OOD.
> The problem of our interest is the case where the data distribution is multimodal, and the problem that the reviewer indicated is another problem of offline RL, which cannot be addressed with our approach.
> As the point the reviewer indicated is a limitation of our work, so we will add a section entitled “Limitation of the proposed method” to the manuscript, and we will discuss when dividing the state action space does not improve the performance of offline RL.
>
> > However, removing components of LAPO may not result in a fair comparison as DMPO also introduces techniques such as advantage normalization in Eq (18).
>
> > It is preferable to see further analysis of why discrete latent variables are better choices than continuous ones, as I found the current empirical comparison versus LAPO- could be made more convincing
>
> We understand the reviewer’s concern. As the reviewer indicated, there were still some differences in the implementation of DMPO and LAPO-. While DMPO employs advantage normalization in Eq. (18), LAPO- employs expectile-regression-like weights for advantage weighting. Thus, we modified the implementation of LAPO- and are running the experiment once again.
>
> The updated version of LAPO- employs the same weights with advantage normalization as AWAC, mixAWAC, and DMPO in our implementation. The network structures of the critics are also the same, and the difference between the updated LAPO-, AWAC, mixAWAC, and DMPO is just the policy structure.
> However, while the policy structure of the updated LAPO- is the same as LAPO, we think that it is not appropriate to call it LAPO- anymore because we made substantial modifications to LAPO for this experiment.
> Therefore, we will refer to the variant of LAPO in Section 7.2 as "LP-AWAC" instead of LAPO-, where "LP" indicates "latent-policy’’.
>
> In addition, we will evaluate LP-AWAC on the toy problem in Section 7.1. To illustrate the difference between LP-AWAC and DMPO.
>
> > ``In addition, DMPO outperformed LAPO-, which is the state-of-the-art method in offline RL." It is arguable that LAPO- can be still considered SOTA when several components of LAPO have been removed.
>
> Thank you for pointing this out. As we modified LAPO substantially, we will remove the above sentence.
>
> We hope that our response will resolve the reviewer's concern. If the above response is not satisfactory, please let us know.

---

> > ### Author Response · Authors · 2023-08-03
> > **Preliminary results on LP-AWAC**
> >
> > As a follow up, we report the results of running LP-AWAC, which employs LAPO-like policy structure based on the continuous latent space and uses the identical network structure and advantage weighting as AWAC and DMPO.
> > We think that the comparison with LP-AWAC will be more convincing than that with LAPO-.
> >
> > The results on locomotion tasks in mujoco-v2 are summarized as follows:
> >
> > ||| LP-AWAC | DMPO | infoDMPO |
> > | --- | --- | ---- | ---- | ---- |
> > |*expert*| HC | 93.7±0.4 | **97.0±1.0** | 95.6±2.0 |
> > |*expert*| HP | **104.3±5.5** | 93.6±15.1 | **107.5±2.9** |
> > |*expert*| WK |110.7±0.1 | 111.4±0.3 | **112.1±0.4** |
> > |*med.-e.*| HC | **92.5±0.4** | **91.1±3.4** | **91.4±2.5** |
> > |*med.-e.*| HP| **90.5±21.6** | 78.4±19.0 | **94.5±14.9** |
> > |*med.-e.*| WK | 109.1±0.4 | **109.9±0.4** | **110.1±0.7** |
> > |*med.-r.*| HC |39.8±0.3 | 45.2±0.8 | **46.7±0.6** |
> > |*med.-r.*| HP| 46.1±8.1 | 89.2±8.1| **98.5±2.0** |
> > |*med.-r.*| WK | 50.2±5.5 | 82.1±3.8 | **86.7±3.2** |
> > |*med.*| HC | 44.0±0.4 | 47.5±0.4 | **48.6±0.4** |
> > |*med.*| HP| 52.8±3.8 | 71.2± 6.5 | **86.4±7.6** |
> > |*med.*| WK | 77.4±2.7 | 79.4±4.7 | **85.0±0.8** |
> >
> > While LP-AWAC demonstrates the best performance on a subset of tasks, overall performance of DMPO and infoDMPO was better than that of LP-AWAC.
> > When the datasets do not contain expert behaviors, the difference between LP-AWAC and DMPO/infoDMPO is clear.
> >
> > The results on the toy problem in Section 7.1 is as follows:
> >
> > |TD3+BC| AWAC | LP-AWAC | DMPO |
> > | --- | --- | ---- | ---- |
> > | -0.2±0.4 | 0.33±0.4 | 0.7±0.6 | 1.0±0.0 |
> >
> > The performance of LP-AWAC is significantly better than TD3+BC and AWAC, demonstrating the benefit of the policy structure based on the latent action space. On the other hand, the performance of DMPO was better than that of LP-AWAC, indicating the advantage of using the discrete latent variable in offline RL.
> >
> > **We found errors in numbers in the above tables when updating the manuscript, so we corrected them. The updated version of the manuscript will be available in a few days. (Aug. 7th)**
> >
> > For now, we are running experiments on antmaze tasks and random mujoco-v2 tasks with the “random” datasets. When the experiments are completed, we will replace the results of LAPO- with LP-AWAC in the manuscript.
> >
> > If you have any questions or comments, please let us know.

---

> > ### Comment · Reviewer_DypM · 2023-08-07
> > **Thank you for the responses**
> >
> > I would like to thank the authors for the responses, additional experiments, and revising the paper. I have a few additional comments and questions regarding the responses.
> >
> > > If the dataset only covers a subset of actions, there is the potential to overestimate the value of actions not contained in the dataset. In such cases, dividing the state action space does not solve the problem of generating OOD.
> >
> > In offline RL, it is often assumed that the dataset only partially covers the state-action space. Otherwise, the offline RL problem is trivialized, as all actions are not OOD. Hence, I believe it is important to elaborate on the advantage of diving state-action space under partial coverage.
> >
> >
> > > We claim that dividing the state-action space can mitigate the problem of generating OOD action when the data distribution in a given dataset is multimodal.
> >
> > Let's take the aforementioned dummy example with an additional action $a_3$ whose reward is $-1$. Suppose the dataset covers $a_1$ and $a_3$. Without using latent policies, the problem of evaluating $a_2$ might still exist regardless the data is unimodal or multimodal.
> >
> > ---
> >
> > I guess the authors' motivation is that dividing state-action space might lead to empirical benefits, by for example (i) making the optimization easier (faster critic loss convergence) or (ii) leading to better mode separation. Please correct me if I am wrong.
> >
> > Demonstrating a fundamental advantage of discrete latent variables over continuous ones is always preferred. However, it would still be interesting if the above empirical advantage could be clearly shown. I understand that the authors aim to demonstrate its empirical advantage by showing (i) the critic loss comparison and (ii) the D4RL benchmark evaluations.
> >
> > I appreciate the authors' effort in making extensive experiments. However, it is a bit difficult for the audience to understand whether the advantage is brought by the discrete latent variables or other factors.
> >
> > As Diffusion-QL [1] was mentioned by reviewer LayE, [1] in fact did a good job in demonstrating the advantage of diffusion policy over other generative models. For example, figure 1 and 2 in [1] shows the effectiveness of Diffusion-QL in mode separation and also in converging to the high-reward mode.
> >
> > Figure 2 in this paper made a similar attempt however it lacks the most important comparison with latent policy with continuous latent variable, LAPO [2]. I guess LAPO could solve the toy example as they show the latent policy could capture the multi-modality in data as shown by Figure 2 in [2].
> >
> > Figures 1 and 2 in [1] could be a good example but there is no need to run the same experiment. Any reasonable toy experiment comparing discrete latent variables versus continuous ones would be appreciated. (My previous point 3.)
> >
> > As the authors mentioned that LP-AWAC will be added to the toy problem in 7.1, I would like to make it clearer that this point will heavily impact my evaluation. The authors may want to ensure the comparison is carefully designed.
> >
> > ---
> > Additional questions/comments:
> >
> > - I wonder why LAPO- degenerates to LP-**AWAC**. According to [2],
> >
> >     - ``Updating the action policy: ... The second step is similar to the advantage-weighted regression (AWR)": the action policy is updated with an AWR style objective.
> >
> >     - ``Policy improvement: LAPO optimizes the latent policy $\pi_\theta(z|s)$ in every iteration to directly maximize the return. It is trained based on standard RL approaches such as DDPG or TD3": the latent policy is updated with a standard policy improvement objective.
> >
> > It seems there are no components that involve AWAC if I was not overlooking.
> >
> > - It would be great if the authors could provide a table showing the differences between LP-AWAC and DMPO, when adding LP-AWAC to the toy experiment.
> >
> > (I'll add an additional message later if I have further questions regarding the preliminary results on LP-AWAC.)
> >
> > ---
> > [1] Wang, Zhendong, Jonathan J. Hunt, and Mingyuan Zhou. "Diffusion Policies as an Expressive Policy Class for Offline Reinforcement Learning." The Eleventh International Conference on Learning Representations. 2022.
> >
> > [2] Chen, Xi, et al. "Lapo: Latent-variable advantage-weighted policy optimization for offline reinforcement learning." Advances in Neural Information Processing Systems 35 (2022): 36902-36913.

---

> > > ### Author Response · Authors · 2023-08-08
> > > **Added the results of LP-AWAC in Section 7**
> > >
> > > Thank you for your constructive suggestions. We revised Section 7 to add the results of LP-AWAC.
> > >
> > > > Without using latent policies, the problem of evaluating $a_2$ might still exist regardless the data is unimodal or multimodal.
> > >
> > > We think that there may be some confusion here. We are not denying the use of the latent policies. Actually, we are using the latent policy in our framework. In our framework, we train a deterministic policy $\mu(s,z)$, which is conditioned on the discrete latent variable $z$. As the obtained latent variable is discrete, learning the latent variable can be considered as dividing the state action space in our framework.
> > >
> > > Regarding the dummy example, our approach would assign different values of the discrete latent variable to $a_1$ and $a_3$, and the mixture of the deterministic policies would try to avoid generating $a_2$, which are not contained in the dataset.
> > >
> > > > As the authors mentioned that LP-AWAC will be added to the toy problem in 7.1, I would like to make it clearer that this point will heavily impact my evaluation. The authors may want to ensure the comparison is carefully designed.
> > >
> > > We added the LP-AWAC in the toy problem in Section 7.1. As Reviewer DypM expected, LP-AWAC solved the problem, demonstrating the benefit of using the latent actor. On the other hand, the performance of DMPO is more stable than that of LP-AWAC, indicating that the use of the discrete latent variable can be advantageous on this toy task.
> > >
> > > In our implementation, the difference between the DMPO and LP-AWAC is the difference between the discrete and continuous latent variables. Both DMPO and LP-AWAC use conditional VAE with advantage weighting to obtain latent variables. The difference is whether conditional VAE learns discrete latent variables or continuous latent variables. The latent actor in LP-AWAC can be considered as the gating policy that approximately solves $\arg \max_{z} Q_w(s,z)$. Therefore, the LP-AWAC can be regarded as the variant of DMPO that employs the continuous latent variable. The results in Sections 7.1 and 7.2 demonstrate the advantage of DMPO over LP-AWAC. Thus, we think that the advantage of the discrete latent variable is now supported by the experimental results.
> > > For implementation details of LP-AWAC, please refer to Appendix F.
> > >
> > > We hope that the additional results of LP-AWAC resolve the concern of Reviewer DypM.
> > >
> > > > I wonder why LAPO- degenerates to LP-AWAC.
> > >
> > > While we modified the implementation of LAPO- to use the advantage normalization in (18) and address the comment from Reviewer DypM, this modification degraded the performance of LAPO-. We think that the expectile-regression-like form of the advantage weight proposed in [2] is also an important technique in LAPO.  To avoid giving a unnecessary negative impression of LAPO to readers, we renamed it.
> > >
> > > Regarding “AWAC” in the name “LP-AWAC”, although the authors of [2] stated ``Updating the action policy: ... The second step is similar to the advantage-weighted regression (AWR)", we think that it is more appropriate to say that the second step is similar to AWAC, not AWR. In AWR, the advantage function is computed based on Monte Carlo return estimate $\mathcal{R}=\sum \gamma^t r_t$, whereas AWAC employs the function approximation for the Q-function. As LAPO employs the approximated Q-function to compute the advantage function, we think that LAPO is similar to AWAC. Thus, we call our varaint LP-AWAC.
> > >
> > >
> > > > It would be great if the authors could provide a table showing the differences between LP-AWAC and DMPO, when adding LP-AWAC to the toy experiment.
> > >
> > > Thank you for the suggestion. We added LP-AWAC to Table 1, which summarizes the difference between baseline and proposed methods.
> > >
> > >
> > > Thanks to the reviewers' constructive comments, we believe that our manuscript has been significantly improved. If any unclear point is remaining, please let us know.

---

> > > > ### Comment · Reviewer_DypM · 2023-08-12
> > > > **Clarification on the latent policy and dummy example.**
> > > >
> > > > Thanks for the response and changes. I would like to clarify on the latent policy and dummy example part.
> > > >
> > > > I understand that DMPO uses latent policy. My point was the latent policy might be the major factor to resolve the problem of evaluating OOD actions, not the discrete latent variables. Therefore I was requesting the comparison between continuous latent variables and discrete ones (LP-AWAC vs. DMPO).
> > > >
> > > > The revised claim
> > > > > Our hypothesis is that, if the data distribution in a given dataset is multimodal, the evaluation of the out-of-distribution actions can be reduced by dividing the state-action space, which can be achieved by learning discrete latent variables of the state-action space.
> > > >
> > > > indicates dividing state-action space alone would address the problem and latent policy is just one potential way to achieve it. It does not show the **necessity** of latent policies.
> > > >
> > > >
> > > > I appreciate the additional experiments and changes in Section 7.1, it resolves my concerns to a good extent. Although I do find the claims do not highlight the necessity of latent policies, it should be fairly easy to fix. Given the additional experiments, I'll revise my review (Claims And Evidence: No -> Yes).

---

> > > > > ### Author Response · Authors · 2023-08-14
> > > > > **Thank you for checking our revision**
> > > > >
> > > > > We appreciate the reviewer's efforts to check our revision, and thank you for the clarification.
> > > > >
> > > > > > indicates dividing state-action space alone would address the problem and latent policy is just one potential way to achieve it. It does not show the necessity of latent policies.
> > > > >
> > > > > We understand what the reviewer means. We will revise the statement according to the comments from the reviewer.

---

### Author Response · Authors · 2023-08-03
**Update log**

We updated the manuscript to address the comments from reviewers. Changes in the manuscript are summarized as follows. We are planning to revise Section 7.2 to address the comments from Reviewer DypM, but it will come later as we are still running the experiments.

- Modified the introduction and added Section 8 “limitation of the proposed method”

To address the comments from Reviewer DypM, we modified a sentence in the introduction:

Before: “Our hypothesis is that the evaluation of the out-of-distribution actions can be avoided by dividing the state-action space, which can be achieved by learning discrete latent variables of the state-action space.”

After: “Our hypothesis is that, if the data distribution in a given dataset is multimodal, the evaluation of the out-of-distribution actions can be reduced by dividing the state-action space, which can be achieved by learning discrete latent variables of the state-action space.”

- Revised the explanation of the prior in Equation 9

We revised the texts to explain the rationale of using the prior in Equation 9 to address the comments from Reviewer LayE.

- Added a paragraph “Approximation gap” on page 5 to discuss the gap between the objective function of AWAC and DMPO

We discussed the approximation gap induced by using the variational lower bound of the log-likelihood to address the comments from Reviewer LayE.

- Revised the description of mutual-information-based regularization on page 5

We corrected the description of MI-based regularization, which was pointed out by Reviewer DypM.

- Revised Section 6 to describe how to model the discrete latent variable

We revised Section 6 to describe how to sample the discrete latent variable to address the comments from Reviewer LayE.

- Added the comparison with Diffusion QL

To address the comments from Reviewer LayE, we added the results on mujoco-v2, antmaze, and adroit tasks. We also added the wall clock time for training and inference.

- Revised a sentence in Section 7.2

We revised a sentence to address the comments from Reviewer DypM

Before: In addition, DMPO outperformed LAPO-, which is the state-of-the-art method in offline RL.

After: In our experiments, DMPO outperformed LAPO-.

Please note that Section 7.2 will be revised as we will replace LAPO- with LP-AWAC later.

- Added Section 8 “Limitation of the proposed method”

To discuss the limitation of the proposed method, we added Section 8 “Limitation of the proposed method.” We described the limitation indicated by Reviewer DypM and discussed the comparison with a diffusion-based policy to address the comments from Reviewer LayE.

- Added how to select the hyperparameter of infoDMPO in Appendix F.

We described how to select the learning rate for maximizing the mutual information term in infoDMPO on page 21.

- Added the implementation details of mixAWAC in Appendix F

We added implementation details of mixAWAC to address the comments from Reviewer LayE.

---

> ### Author Response · Authors · 2023-08-03
> **Implementation of mixAWAC and DMPO**
>
> Regarding our implementation, we are planning to make the codes of AWAC, mixAWAC, DMPO, and infoDMPO public if the paper gets accepted. We think that the details of mixAWAC and proposed methods can be found when the codes are public.

---

> > ### Author Response · Authors · 2023-08-23
> > **Update of the introduction**
> >
> > To address the comments from Reviewer DypM, we revised a sentence in the introduction:
> >
> > Before: "Our hypothesis is that, if the data distribution in a given dataset is multimodal, the evaluation of the out-of-distribution actions can be reduced by dividing the state-action space, which can be achieved by learning discrete latent variables of the state-action space."
> >
> > After: "Our hypothesis is that, if the data distribution in a given dataset is multimodal, the evaluation of the out-of-distribution actions can be reduced by leveraging a policy conditioned on discrete latent variables, which can be interpreted as dividing the state-action space and learning sub-policies for each region."

---

### Decision · Action_Editors · 2023-08-29

**Recommendation:** Accept with minor revision

**Comment:**

In this paper, the authors identified an issue in the major existing offline RL methods, which is the multimodality in policy. To overcome the problem, the authors introduced the deterministic mixture model for policy parametrization, which will be learned by a variational lower bound of the policy value, with a mutual-information regularization.

The authors justified the performance of the proposed method on the existing benchmarks, demonstrating their benefits. After the rebuttal discussion period, the authors added more comparison on LAPO and diffusion-QL, which are the major competitors and should be added originally.

In sum, all the reviewers acknowledge that although the idea to use more flexible policy parametrization is already proposed for offline RL, this paper is still solid. Considering the novelty is not the major evaluation criterion, the paper should be accepted.

There are several minors need to be addressed:

1, The gradient approximation for discrete variable in the loss function is not specified. This part is very important for reproducibility. Please add the details of the implementations about gradient calculation.

2, The loss function is confusing in Eq 6. I guess it should be
$\int d(s)\beta(s, a)\log (\frac{\pi(a|s)}{\beta(a|s)}) f(s, a)dsda$ instead of $\int d(s)\beta(s, a)\log (\frac{\pi(a|s)}{\beta(a|s)} f(s, a) )dsda$? Please make this point clear.

**Audience:**

The major audience of this submission is the offline RL community.

**Claims And Evidence:**

As all reviewers agreed, this paper is well-organized and easy to follow.

The limitation of the proposed method is clearly stated in the paper.

---

> ### Author Response · Authors · 2023-08-31
> **Thank you for evaluating our manuscript**
>
> We appreciate the efforts of  AE and reviewers for reviewing our manuscript. Our manuscript has been significantly improved by constructive comments from AE and reviewers.
>
> > 1, The gradient approximation for discrete variables in the loss function is not specified. This part is very important for reproducibility. Please add the details of the implementations about gradient calculation.
>
> We used the  Gumbel-max trick to sample the discrete variable in a differentiable manner. As we will include the link to our codes, the details should be clear in the camera-ready version.
>
> > 2, The loss function is confusing in Eq 6. I guess it should be $\iint d(s)\beta(a|s)\log ( \frac{\pi(a|s)}{\beta(a|s)}) f(s,a) dads$ instead of $\iint d(s)\beta(a|s)\log ( \frac{\pi(a|s)}{\beta(a|s)} f(s,a) )dads$ ? Please make this point clear.
>
> Thank you for pointing this out. AE is right. We will revise the equation accordingly for the camera-ready version.